# IL-22 initiates an IL-18-dependent epithelial response circuit to enforce intestinal host defence

Hung-Yu Chiang [1], Hsueh-Han Lu [1], Janaki N. Sudhakar [1], Yu-Wen Chen [1,2], Nien-Shin Shih[1], Yi-Ting Weng[1] & Jr-Wen Shui [1]✉

IL-18 is emerging as an IL-22-induced and epithelium-derived cytokine which contributes to host defence against intestinal infection and inflammation. In contrast to its known role in Goblet cells, regulation of barrier function at the molecular level by IL-18 is much less explored. Here we show that IL-18 is a bona fide IL-22-regulated gate keeper for intestinal epithelial barrier. IL-22 promotes crypt immunity both via induction of phospho-Stat3 binding to the *Il-18* gene promoter and via *Il-18* independent mechanisms. In organoid culture, while IL-22 primarily increases organoid size and inhibits expression of stem cell genes, IL-18 preferentially promotes organoid budding and induces signature genes of Lgr5+ stem cells via Akt-Tcf4 signalling. During adherent-invasive *E. coli* (AIEC) infection, systemic administration of IL-18 corrects compromised T-cell IFNγ production and restores Lysozyme+ Paneth cells in *Il-22*−/− mice, but IL-22 administration fails to restore these parameters in *Il-18*−/− mice, thereby placing IL-22-Stat3 signalling upstream of the IL-18-mediated barrier defence function. IL-18 in return regulates Stat3-mediated anti-microbial response in Paneth cells, Akt-Tcf4-triggered expansion of Lgr5+ stem cells to facilitate tissue repair, and AIEC clearance by promoting IFNγ+ T cells.

---

[1] Institute of Biomedical Sciences, Academia Sinica, Taipei, Taiwan. [2] Taiwan International Graduate Program in Molecular Medicine, National Yang Ming Chiao Tung University and Academia Sinica, Taipei, Taiwan. ✉email: jshui@ibms.sinica.edu.tw

Impaired host defence to intestinal pathogens has been implicated to associate with increased susceptibility to inflammatory disorders including inflammatory bowel disease (IBD)[1]. Certain Gram-negative bacteria, such as *Salmonella typhimurium*, *Yersinia enterocolitica*, or *Adherent-invasive Escherichia coli* (AIEC), are known to cause intestinal symptoms. Among them, Yersinia and AIEC are described as a pathogenic trigger for IBD and high prevalence of AIEC in patients with Crohn's disease (CD) has been reported[2–4]. In animal models, persistent infection with AIEC in the gut is associated with chronic inflammation and fibrosis, which CD8+ or IFNγ-producing immune cells are shown to be protective[5]. While Th1 and Th17-mediated immune responses are involved in the pathogenesis of Crohn's disease and in AIEC-infected mice[5,6], convincing genetic evidence are still needed to elucidate how the frontline epithelial cells and their corresponding biological products, such as cytokines or anti-microbial peptides, are involved in AIEC host defence.

IL-22 is an ILC3 (innate lymphoid cell type 3) or Th17 signature cytokine and is generally known as a tissue protective cytokine which exclusively targets epithelial lineages[7,8]. IL-22 has a key role in epithelial barrier mainly because of its capability to induce epithelial production of anti-microbial and anti-inflammatory mediators, as well as to promote epithelial regeneration or wound healing[8]. As such, both ILC3 and Th17 cells are implicated in IBD pathogenesis[6,7]. Clinical relevance of IL-22 to IBD is well-established and IL-22 therapy is considered as a promising strategy for IBD treatment[8,9]. In IL-22-mediated signalling cascade, IL-18 is recently emerging as an important IL-22-induced and epithelium-derived cytokine, whose functionality is completely different from those established roles for IL-18 in inflammasome in myeloid cells[10]. For example, epithelial IL-18 not only relays IL-22-IL-22R signalling for host defence but is also required for IL-22 production during intestinal helminth infection[11]. Both protective and detrimental functions of IL-18 in the gut have been reported. Epithelial inflammasome-derived IL-18 was shown to activate an anti-microbial program which subsequently regulates microbial community to prevent intestinal inflammation[12]. A recent study reports that IL-22 and IL-18 promote intestinal epithelial migration for a rapid turnover leading to the protection against rotavirus infection[13]. In contrast, epithelial IL-18-IL-18R signalling is described to inhibit Goblet cell maturation which consequently might exacerbate colitis[14]. Regarding Paneth cells, while one study reports that IL-22-stimulated organoids do not lead to enhanced Paneth cell frequency or Stat3 phosphorylation[15], new evidence show that IL-22 promotes the production and secretion of Lysozyme, or the induction of specific markers (Lysozyme, MMP7), from Paneth cells in organoids[16,17]. In addition, IL-22R signalling in Paneth cells contributes to their maturation and full host protection against Salmonella infection[18]. Nevertheless, a role for IL-18 in Paneth cells or epithelial stem cells has not been studied. It is known that, in the milieu of IL-12, IL-18 is a potent IFNγ inducer in Th1 cells, suggesting that the IL-18-IFNγ axis could potentially contribute to the pathogenesis of Crohn's disease[19]. Supporting this notion, levels of IL-18 in serum or mucosal epithelial biopsies were significantly higher in CD patients and *Il-18* gene polymorphism is linked to increased susceptibility to Crohn's disease[20,21]. While the interplay between IL-22 and IL-18 appears cross-regulated during inflammation or host defence, evidence of a specific role for IL-18 in epithelial barrier, with respect to what are definitive target cell types of IL-18 during CD-related host defence, is missing.

Here we show, using a clinical isolate of AIEC from CD patients, a coordinated response of IL-22 and IL-18 to intestinal AIEC infection. While these two cytokines show integrated functions in regulating Paneth cells and Goblet cells, they surprisingly exert distinct regulatory functions and pathways towards epithelial stem cells. At the molecular level, we identify the requirement of Stat3 in epithelial transcriptional induction of IL-18 by IL-22, of Akt-Tcf4 in IL-18-mediated transcriptional activation of Lgr5, and of Stat3 in IL-22/IL-18-induced Paneth cell functionality. A full coordinated IL-22-initiated IL-18 response circuit to enforce mucosal host defence against Crohn's AIEC is proposed and discussed.

## Results

### AIEC infection triggers an early response of stem cells and Paneth cells.
Pathogenic microbes in the intestine have been implicated as a trigger for IBD[4,22]. Adherent-invasive *E. coli* (AIEC) is abundantly identified in the ileum mucosa of CD patients and a clinical isolate of AIEC has been shown to cause chronic inflammation and fibrosis in mice[3,5]. As functionality of stem cell-mediated epithelial regeneration and Paneth cell-directed anti-microbial peptide (AMP) production is critical for epithelial barrier against mucosal infection[23], we asked how Crohn's AIEC initiates the immune response of these two frontline cell types. As noted, oral gavage with an AIEC isolate NRG857c caused a rapid epithelial apoptosis in ileal epithelium[5], marked by an increase of active Caspase-3+ crypts at the early stage of infection (Fig. 1a, b), and a corresponding epithelial regeneration evidenced by an increase of CD24$^{-/low}$ Ki67+ proliferating crypts which mostly represent transit-amplifying (TA) cells (Figs. 1c, 2b)[24,25]. Correlated to TA cell response, the stem cell response, marked by an increase of CD24$^{-/low}$ Lgr5+ crypts[24], or mRNA expression of stem cell marker Lgr5, Ascl2, and Olfm4[26], was also upregulated at the early stage (Fig. 1d-f and Supplementary Fig. 1a-b). This indicates that epithelial regeneration is also rapidly and actively triggered as a repair host defence against invasive and destructive property of AIEC. Furthermore, the response of Paneth cells (PC), which was marked by the missing CD24+ Lysozyme+ subset in ileum crypts of Paneth cell-deficient (*Defa6-Cre+Rosa26-LSL-DTA* or PC$^\Delta$) mice and by the PC marker Lysozyme or Cryptdin (Fig. 1g-h and Supplementary Fig. 1a, c)[24,27], was also actively triggered at the early stage. As noted, PC-deficient mice were more susceptible to AIEC infection (Fig. 1i). Therefore, mucosal infection of Crohn's AIEC causes an early and robust response program of epithelial stem cells and Paneth cells, whose full functionality might be important to prevent chronic inflammation or fibrosis[5].

### IL-22-Stat3 axis upregulates Paneth cells in response to AIEC.
IL-22 is associated with IBD because of its capability to promote regeneration and anti-microbial function of epithelial barrier[8]. As such, we asked how IL-22 is involved in AIEC host defence. As noted, IL-22 is abundant in the lamina propria (LP) compartments (Supplementary Fig. 1d). At the early stage of infection, IL-22 was rapidly induced in the ileum LP (Fig. 2a), indicating that IL-22 initiates an innate response to AIEC which is known to target the ileum. Next, at the steady state or during AIEC infection, loss of *Il-22* in mice caused a significant decrease in Ki67+ proliferating TA cells (Fig. 2b-c), which is consistent to previous studies where injection of IL-22 into mice increases epithelial proliferation and expands TA compartments[15,17]. Of note, Ki67+ proliferating crypts are also reduced in epithelium-specific *Stat3* conditional knockout (*Vil-Cre+Stat3f/f*) mice[28], indicative of a role for IL-22-Stat3 signalling in epithelial regeneration during mucosal infection. We next tested the controversial role of IL-22 in the regulation of Paneth cells. While one study shows undetectable IL-22R expression and IL-22-induced Stat3 phosphorylation in Paneth cells[15], a recent study provides genetic evidence that IL-22R signalling in Paneth cells is required for maturation and full protection against *Salmonella typhimurium*[18]. As noted, loss of *Il-22* or epithelial *Stat3* in mice significantly compromised

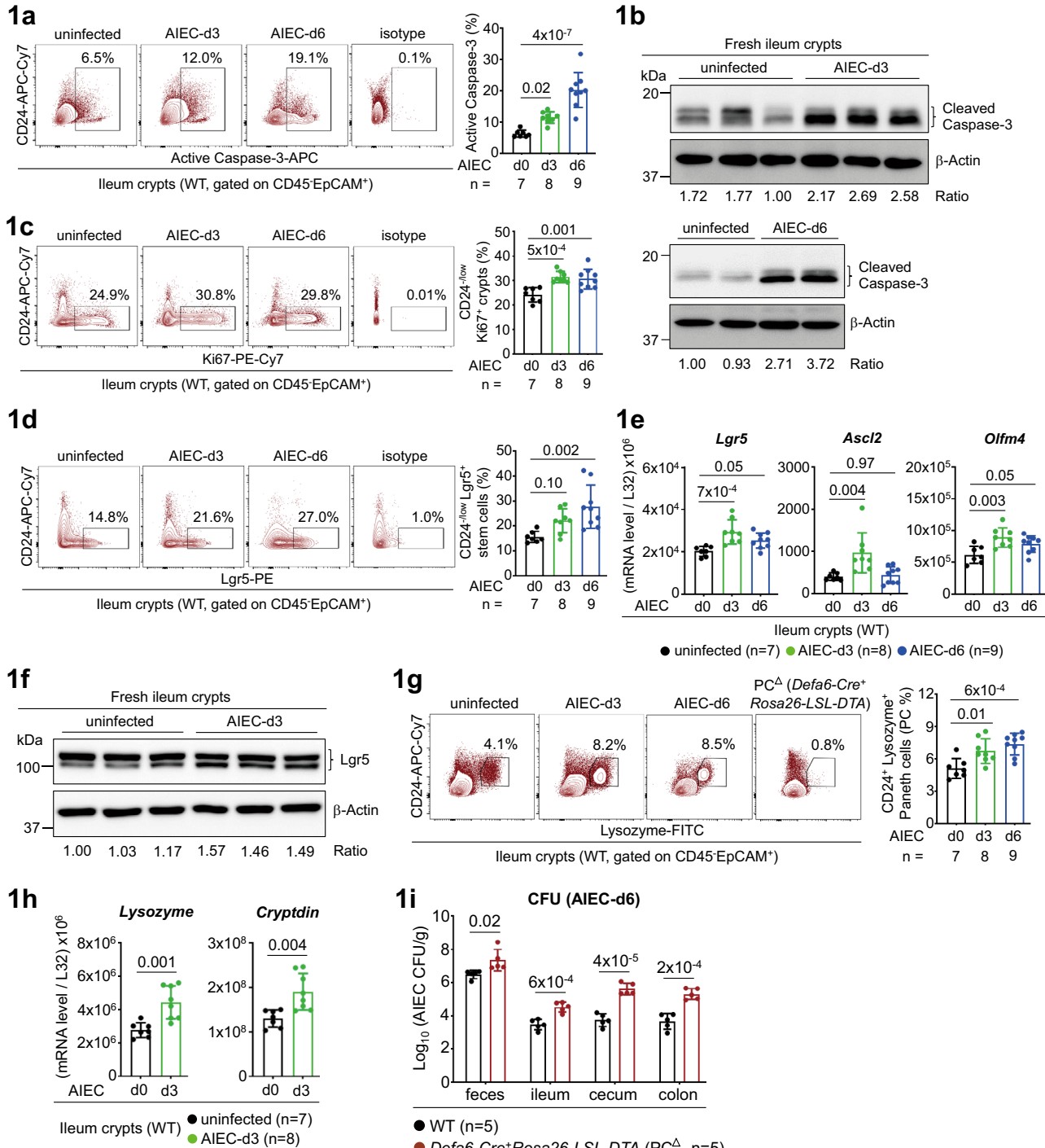

**Fig. 1 Crohn's AIEC induces an early response of stem cells and Paneth cells. a** Flow cytometry analysis of CD45$^-$ EpCAM$^+$ ileum crypts for active Caspase-3$^+$ apoptotic cells in uninfected and AIEC-infected mice at day-3 (d3) and day-6 (d6) post infection. **b** Western blot analysis of ileum crypts for active Caspase-3$^+$ apoptotic cells. Quantification and the ratio (Caspase-3 to β-actin) of protein bands are indicated. **c** Flow cytometry analysis of CD24$^{-/low}$ ileum crypts for Ki67$^+$ proliferating cells in uninfected and AIEC-infected mice at d3 and d6. **d** Flow cytometry analysis of CD24$^{-/low}$ ileum crypts for Lgr5$^+$ stem cells in uninfected and AIEC-infected mice at d3 and d6. **e** Quantitative real-time PCR analysis of ileum crypts for stem cell marker Lgr5, Ascl2, and Olfm4 in uninfected and AIEC-infected mice. **f** Western blot analysis of ileum crypts for stem cell marker Lgr5 in the indicated mice. Quantification and the ratio (Lgr5 to β-actin) of protein bands are indicated. **g** Flow cytometry analysis of CD24$^+$ ileum crypts for Lysozyme$^+$ Paneth cells in the indicated mice. Ileum crypts isolated from Paneth cell-deficient (PC$^\Delta$) Defa6-Cre$^+$Rosa26-LSL-DTA mice were used to gate the Paneth cell population. **h** Quantitative real-time PCR analysis of ileum crypts for Paneth cell marker Lysozyme and Cryptdin in the indicated mice. **i** Colony-forming unit (CFU) of AIEC was analyzed in feces and intestinal tissues isolated from the indicated mice at d6 post AIEC infection. Each symbol in bar graphs represents an ileum crypt sample (**a**, **c**, **d**, **e**, **g**, **h**) or tissue sample (**i**) derived from one mouse. Data shown are representative (**b**, **f**, **i**) or combined (**a**, **c**, **d**, **e**, **g**, **h**) results from two independent reproducible experiments. Statistical significance is indicated using unpaired two-tailed *t* test (**h**, **i**) or One-way ANOVA with Sidak's multiple comparisons test (**a**, **c**, **d**, **e**, **g**). Data are presented as mean ± SD. Source data are provided as a Source Data file.

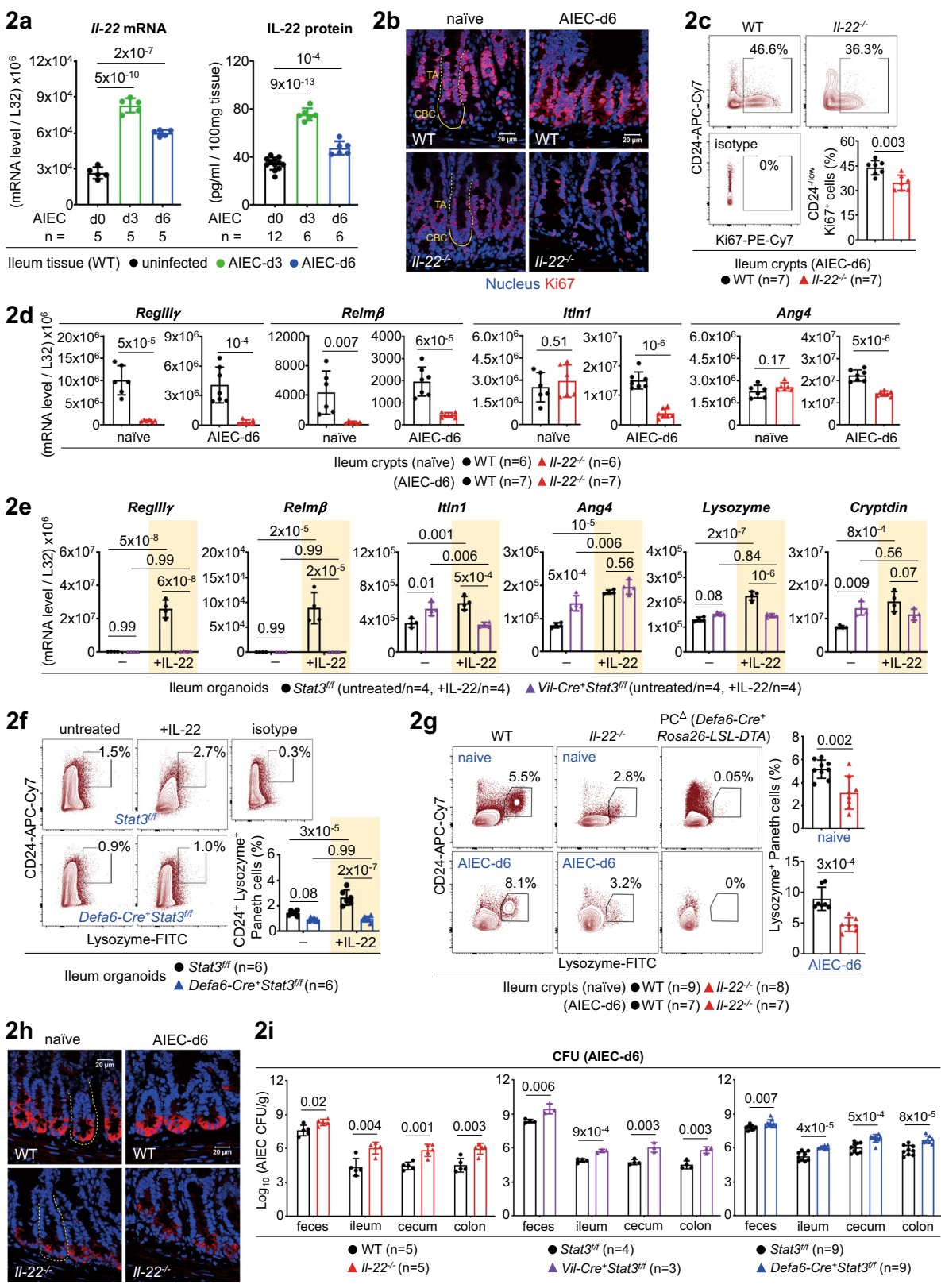

the AMP production (Fig.2 d, and Supplementary Fig. 1e). Using ileum organoids derived from *Vil-Cre⁺Stat3^f/f* (epithelium-specific) or *Defa6-Cre⁺Stat3^f/f* (Paneth cell-specific) mice as a gain-of-function approach, we found that IL-22 upregulates non-Paneth cell AMP (RegIIIγ, Relmβ), Paneth cell-related AMP (Itln1, Ang4), and Paneth cell-specific AMP (Lysozyme,

Cryptdin) all in a Stat3-dependent manner (Fig. 2e-f, and Supplementary Fig. 1f-g). To further validate a regulatory role for IL-22 in Paneth cell function, loss-of-function studies by flow cytometry, immunofluorescence, and quantitative PCR analyses of Paneth cells in ileum tissues of steady-state or AIEC-infected *Il-22⁻/⁻* mice were tested (Fig. 2g-h, and Supplementary Fig. 1h).

**Fig. 2 IL-22-Stat3 axis upregulates Paneth cells for AIEC host defence. a** Quantitative real-time PCR and ELISA analyses of ileum lamina propria cells for IL-22 in uninfected or AIEC-infected mice. **b** Immunofluorescence analysis of ileum crypts for Ki67$^+$ proliferating cells in the indicated mice. Crypt base columnar (CBC, outlined with a solid line) stem cell and the above transit-amplifying (TA, outlined with a dashed line) compartments are indicated. **c** Flow cytometry analysis of CD24$^{-/low}$ ileum crypts for Ki67$^+$ proliferating cells in AIEC-infected mice at day-6. **d** Quantitative real-time PCR analysis of ileum crypts for anti-microbial peptides in uninfected or AIEC-infected *Il-22$^{-/-}$* mice. **e** Quantitative real-time PCR analysis of IL-22-stimulated ileum organoids, derived from the indicated mice, for anti-microbial peptides. **f** Flow cytometry analysis of IL-22-stimulated ileum organoids, derived from the indicated mice, for CD24$^+$ Lysozyme$^+$ Paneth cells. **g** Flow cytometry analysis of ileum crypts, derived from the indicated mice, for CD24$^+$ Lysozyme$^+$ Paneth cells. **h** Immunofluorescence analysis of ileum crypts (outlined with a dashed line) for Lysozyme$^+$ Paneth cells at the crypt base in the indicated mice. **i** Colony-forming unit (CFU) of AIEC was analyzed in feces and intestinal tissues isolated from the indicated mice at day-6 (d6) post infection. Each symbol in bar graphs represents an ileum crypt sample (**c**, **d**, **g**), tissue sample (**a**, **i**), or organoid culture (**e**, **f**), derived from one mouse. Data shown are representative (**b**, **h**) or combined (**a**, **c**, **d**, **e**, **f**, **g**, **i**) results from two independent reproducible experiments. Figures 2f and 4c were performd in the same experiment. Statistical significance is indicated using unpaired two-tailed *t* test (**c**, **d**, **g**, **i**), One-way ANOVA with Sidak's multiple comparisons test (**a**), or Two-way ANOVA with Tukey's multiple comparisons test (**e**, **f**). Data are presented as mean ± SD. Source data are provided as a Source Data file.

Furthermore, a crucial role for IL-22-Stat3 signalling in crypt regeneration and Paneth cells during host defence was determined by showing increased susceptibility of *Il-22$^{-/-}$*, *Vil-Cre$^+$Stat3$^{f/f}$*, and *Defa6-Cre$^+$Stat3$^{f/f}$* mice to AIEC infection (Fig. 2i). Taken together, our results support that IL-22 is a bona fide regulator of Paneth cells both in vitro and in vivo. We conclude that in vivo maintenance of Ki67$^+$ crypt regeneration and Paneth cells requires IL-22-Stat3 signalling and AIEC infection triggers an epithelial IL-22-Stat3-mediated Paneth cell response.

**IL-22-Stat3 axis transcriptionally induces epithelial IL-18.** While IL-22-mediated crypt regeneration and Paneth cell response provide the first line of AIEC host defence at the early stage, IFNγ and CD8$^+$ T cells were shown to be crucial at the later stage of infection[5]. We then asked whether IFNγ is important for AIEC clearance in our platform and whether IL-22 is linked to the upregulation of IFNγ in T cells during AIEC infection. Notably, *Ifnγ$^{-/-}$* mice showed increased susceptibility to AIEC infection (Fig. 3a) and analysis of ileum lamina propria in AIEC-infected *Il-22$^{-/-}$* mice showed an over 50% reduction of IFNγ-producing CD8$^+$ T cells (Fig. 3b, Supplementary Fig. 2a), indicating that IL-22 is functionally linked to the induction of IFNγ$^+$ CD8$^+$ T cells during AIEC infection. Among potential upstream regulators of IFNγ, IL-18, in the presence of IL-12, is a potent IFNγ inducer in CD8$^+$ T cells[29]. Indeed, we found that IL-18 and IL-12 robustly and preferentially promote IFNγ$^+$ CD8$^+$ T cells in mesenteric lymphocytes (Supplementary Fig. 2b–d), raising the possibility that IL-18 might relay IL-22 signalling to downstream IFNγ response for AIEC host defence. Supporting this notion, IL-22 has been shown to maintain epithelial homeostasis of IL-18 and the IL-22-IL-18 axis contributes to host defence against helminth[11,29]. As noted, IL-18 expression in *Il-22$^{-/-}$* ileum crypts was reduced at the steady state or during AIEC infection (Fig. 3c-d). Furthermore, as there was no difference in IL-12 expression in ileum lamina propria between wild-type and *Il-22$^{-/-}$* mice (Supplementary Fig. 2e), it is likely that IL-22, but not IL-12, upregulates IL-18 leading to IFNγ response. We also found that epithelium-specific *Stat3* conditional deletion in mice also causes a decrease of IL-18 in ileum crypts (Fig. 3c), suggesting that the IL-22-Stat3 signalling is linked to downstream IL-18-IFNγ cascade. To gain more insights, ileum organoids were in vitro stimulated with recombinant IL-22 or IL-18, and analyzed for IL-18 upregulation. The results showed that IL-22 induces IL-18 in a Stat3-dependent manner but IL-18 itself does not induce IL-18 (Fig. 3e–f). As noted, epithelial *Stat3* deletion caused more reduction of IL-18 after IL-22 stimulation, compared to Paneth-cell *Stat3* deletion. This might be due to the fact that IL-22R is more expressed in CD24$^{low}$ Lysozyme$^-$ epithelial subset than in Lysozyme$^+$ Paneth cells (Fig. 3g, Supplementary Fig. 2f).

As Stat3 is a transcription factor, we next asked whether IL-22 regulates Stat3 binding to the *Il-18* promoter for transcriptional upregulation. To this end, we analyzed and identified five (marked as P1~P5) STAT consensus binding sites within the mouse *Il-18* promoter (Supplementary Fig. 3a). Next, chromatin immunoprecipitation (ChIP) assay, with a specific anti-phospho-Stat3$^{Tyr705}$ (active form of Stat3) antibody for immunoprecipitation, was performed in IL-22-stimulated ileum crypts, to determine whether and where IL-22 promotes phospho-Stat3 binding to those putative STAT binding sites (Supplementary Fig. 3b). As noted, we found that IL-22 induces phospho-Stat3 binding to the P3, P4, and P5 sites within the *Il-18* promoter (Fig. 3h). We also analyzed the human *Il-18* promoter and performed the ChIP assay in human colon epithelial HT-29 cells. The result showed that IL-22 induces phospho-Stat3 binding to the P2 site within the human *Il-18* promoter (Supplementary Fig. 3c–d). Collectively, our results reveal an IL-22-initiated IL-18 response cascade, where IL-22 transcriptionally activates epithelial IL-18 by promoting phospho-Stat3 binding to the *Il-18* promoter, leading to subsequent IL-12/IL-18-mediated IFNγ induction preferentially in CD8$^+$ T cells which are crucial for AIEC clearance.

**IL-18 is a bona fide regulator of Paneth cells for AIEC infection.** In addition to a role for IL-18 in IFNγ induction, we asked how IL-18 regulates epithelial barrier during AIEC host defence. As noted, different from IL-22, IL-18 is more enriched in the crypt compartments (Supplementary Fig. 4a), suggesting that epithelium-derived IL-18 might have a more important role over myeloid cell-derived IL-18 which is enriched in the lamina propria. Similar to an early induction of IL-22 in ileum lamina propria, IL-18 was also rapidly upregulated in ileum crypts at the early stage (Fig. 4a). The same induction kinetics provides a functional link that IL-22-initiated IL-18 is triggered as an innate response. Notably, in ileum organoids, both AIEC and IL-22 can directly induce IL-18 (Supplementary Fig. 4b). IL-18, like IL-22, is also capable of inducing Paneth cell-related AMP (Itln1, Ang4) and Paneth cell-specific AMP (Lysozyme, Cryptdin), all in a Stat3-dependent manner (Fig. 4b-c, Supplementary Fig. 4c-d). Intriguingly, while both IL-22 and IL-18 can induce Stat3 phosphorylation in epithelial cells including Lysozyme$^+$ Paneth cells and Muc2$^+$ Goblet cells (Fig. 4d, Supplementary Fig. 4e), IL-18R is predominantly expressed in CD24$^{high}$ Lysozyme$^+$ Paneth cells (Fig. 3g), compared to IL-22R in the CD24$^{low}$ Lysozyme$^-$ subset. The differential expression pattern of IL-22R and IL-18R raises the possibility that IL-22 and IL-18 may target the same or different subsets for redundant and non-redundant functionality. Indeed, while IL-18 is incapable of inducing certain IL-22-directed non-Paneth cell AMP (RegIIIγ, Relmβ), IL-22/IL-18-stimulated crypts exert an additive effect on bacterial killing (Fig. 4e, Supplementary Fig. 4c)[30]. Next, we further explored how

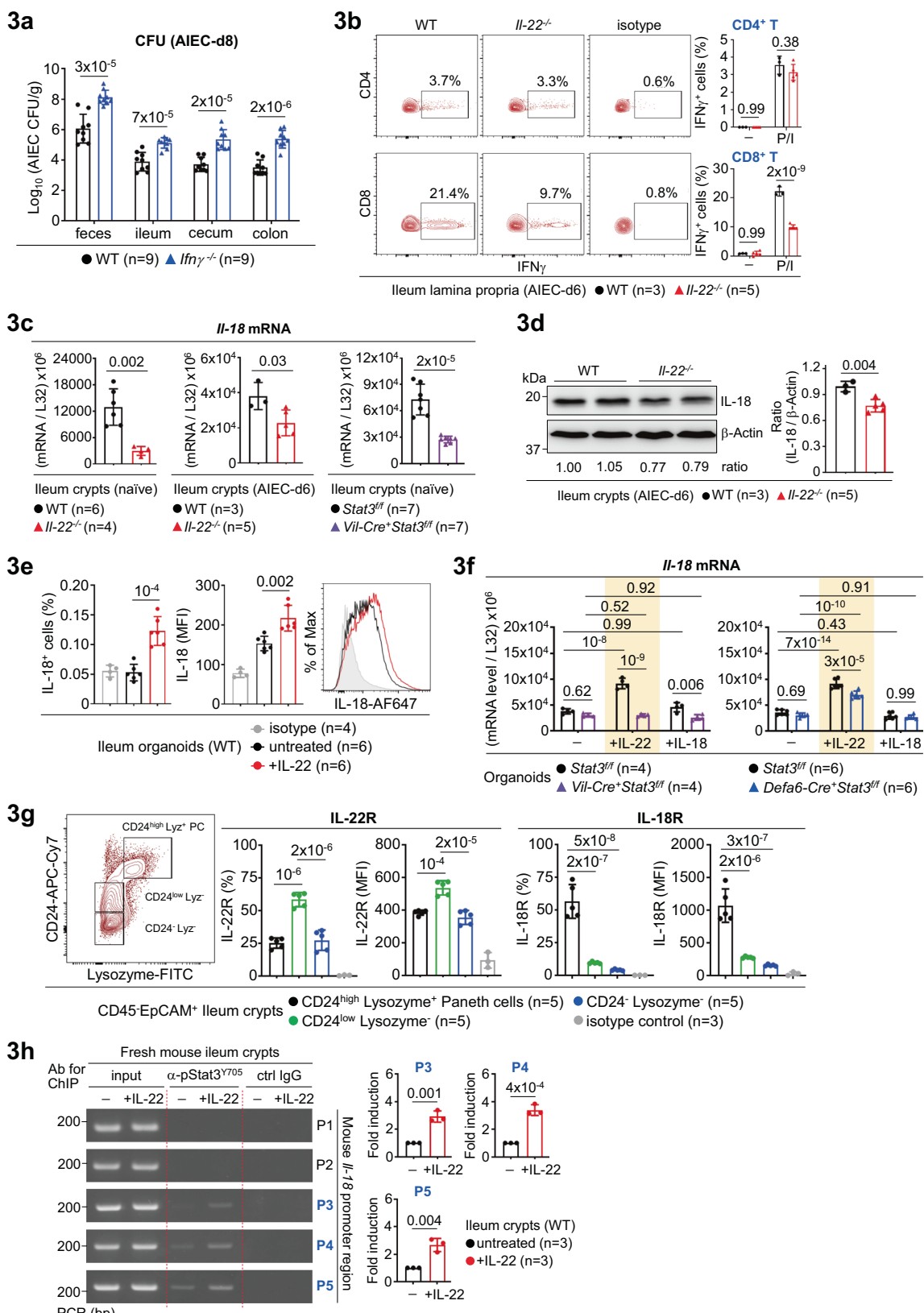

IL-18 contributes to Paneth cell homeostasis. As noted, loss of *Il-18* in mice led to reduced mRNA levels of AMP, IFNγ, Dll1/4 (Notch ligands), and Hes1 (Notch target genes)[16], but not Atoh1 (Supplementary Fig. 4f-g)[27], a key transcription factor important for the differentiation of epithelial secretory precursor. By immunofluorescence, flow cytometry, H&E staining, and quantitative PCR analyses, Lysozyme-expressing and granule-containing Paneth cells were significantly reduced in *Il-18*[−/−]

**Fig. 3 IL-22 transcriptionally induces epithelial IL-18 via phospho-Stat3$^{Y705}$. a** Colony-forming unit (CFU) of AIEC was analyzed in feces and intestinal tissues isolated from the indicated mice at day-8 (d8) post infection. **b** Flow cytometry analysis of PMA/Ionomycin (P/I)-stimulated ileum lamina propria cells for intracellular IFNγ in CD4$^+$ or CD8$^+$ T cells in AIEC-infected mice. **c** Quantitative real-time PCR analysis of ileum crypts from the indicated mice for *Il-18* mRNA levels. **d** Western blot analysis of ileum crypts from the indicated mice for IL-18. Quantification and the ratio (IL-18 to β-actin) of protein bands are indicated. **e** Flow cytometry analysis of IL-22-stimulated ileum organoids for intracellular IL-18. **f** Quantitative real-time PCR analysis of IL-22 or IL-18-stimulated ileum organoids, derived from the indicated mice, for *Il-18* mRNA levels. **g** Flow cytometry analysis of naïve wild-type ileum crypts for IL-22R or IL-18R expression levels in the indicated epithelial subsets. **h** Chromatin immunoprecipitation (ChIP) and PCR analyses of IL-22-stimulated ileum crypts for the binding of phospho-Stat3$^{Y705}$ to the mouse *Il-18* promoter regions (P1-P5). Fold induction is calculated based on PCR signal strength of pStat3-bound *Il-18* promoter region (P3,P4,P5) before and after IL-22 stimulation. Each symbol in bar graphs represents an ileum crypt sample (**c**, **d**, **g**, **h**), tissue sample (**a**, **b**), or organoid culture (**e**, **f**), derived from one mouse. Data shown are representative (**b**, **d**, **g**, **h**) or combined (**a**, **c**, **e**, **f**) results from two independent reproducible experiments. Statistical significance is indicated using unpaired two-tailed *t* test (**a**, **c**, **d**, **e**, **h**), One-way ANOVA with Sidak's multiple comparisons test (**g**), or Two-way ANOVA with Tukey's multiple comparisons test (**b**, **f**). Data are presented as mean ± SD. Source data are provided as a Source Data file.

crypts at the steady state or during AIEC infection (Fig. 4f-g, Supplementary Fig. 4h-i), indicative of a crucial role for IL-18 in Paneth cell maintenance. As noted, while loss of *Il-18* at the steady state did not cause detectable apoptosis in Paneth cells located at the crypt base, Caspase-3$^+$ Paneth cells were increased during AIEC infection (Supplementary Fig. 4j-k), supporting a role for IL-18 in maintaining Paneth cells during host defence. Correlated to reduced AMP response and Paneth cell dysfunction in the absence of IL-18, *Il-18$^{-/-}$* mice were indeed more susceptible to AIEC infection with impaired bacterial clearance (Fig. 4h). As noted, the defective host defence in *Il-18$^{-/-}$* mice was also attributed by impaired T-cell induction of IFNγ in the lamina propria (Fig. 4i, Supplementary Fig. 4g). To validate the key role of IL-18-Stat3 signalling in Paneth cells during AIEC infection, we investigated how the injection of recombinant IL-18 into *Defa6-Cre$^+$Stat3$^{f/f}$* mice contributes to AIEC host defence (Supplementary Fig. 6a). Intriguingly, exogenous IL-18 failed to rescue the susceptibility of *Defa6-Cre$^+$Stat3$^{f/f}$* mice to AIEC infection, as evidenced by higher bacterial burdens and compromised antimicrobial response of Paneth cells after IL-18 administration (Fig. 4j-m, Supplementary Fig. 4l). This indicates that IL-18-Stat3 signalling in Paneth cells is crucial for AIEC host defence. To further provide evidence that IL-18 indeed directly target Paneth cells, the EpCAM$^+$ CD24$^{+/high}$ c-Kit$^+$ CD66a$^{-/low}$ crypt subset was tested positive as Lysozyme-producing Paneth cells and then sorted for IL-18 stimulation (Supplementary Fig. 5a)[31]. Unlike ileum fragments or fresh crypts, we found that only organoids and sorted Paneth cells respond well to IL-18 treatment by inducing Paneth cell-specific AMP (Lysozyme and Cryptdin) and Paneth cell-related AMP (Itln1, Ang4, MMP7) (Supplementary Fig. 5b-c). Collectively, our gain-of-function study in IL-18-stimulated organoids or sorted Paneth cells and in IL-18-injected mice, as well as loss-of-function study in *Il-18$^{-/-}$* mice, confirm that IL-18 is a bona fide regulator of Paneth cells both in vitro and in vivo.

**IL-22 links IL-18 for IFNγ and Paneth cell responses**. To provide in vivo evidence that IL-22 links IL-18 to upregulate T-cell production of IFNγ in response to AIEC infection, we first investigated how the injection of recombinant IL-18 into *Il-22$^{-/-}$* mice contributes to host defence. As noted, exogenous IL-18 effectively reduced bacterial burdens in most of intestinal tissues in infected *Il-22$^{-/-}$* mice (Fig. 5a, Supplementary Fig. 6a), indicating that IL-18 administration can rescue or bypass the defect of host defence caused by IL-22 deficiency. Correlated to IL-18-mediated CFU reduction, IL-18 injection upregulated IFNγ production in both CD4$^+$ and CD8$^+$ T cells in the ileum lamina propria of *Il-22$^{-/-}$* mice (Fig. 5b), indicative of the importance of IL-22-IL-18-IFNγ axis in host defence. Furthermore, it appears that IL-18-mediated restoration of Lysozyme$^+$ Paneth cells also

contributes to improved host defence in *Il-22$^{-/-}$* mice (Fig. 5c). Intriguingly, in reciprocal rescue experiments, IL-22 injection into *Il-18$^{-/-}$* mice failed to improve bacterial clearance, as well as T-cell production of IFNγ in the lamina propria (Fig. 5d–e). Unexpectedly, while IL-22 itself can promote Paneth cells in organoid culture, it failed to restore Lysozyme$^+$ Paneth cells in *Il-18$^{-/-}$* mice during AIEC infection (Fig. 5f). This indicates that IL-22 regulates IL-18-dependent Paneth cell functionality. It is likely that IL-22R expression levels in the context of AIEC infection is not sufficient to upregulate Paneth cells, such that IL-22-induced IL-18 plays a predominant role because of high IL-18R expression in Paneth cells (Fig. 3g). Correlated to the in vivo findings, we found that in *Il-18$^{-/-}$* organoids, while IL-22 still induces certain non-Paneth cell AMP (RegIIIγ, Relmβ) and Paneth cell-related AMP (Itln1, Ang4), IL-22 indeed fail to induce Paneth cell-specific AMP (Lysozyme, Cryptdin) (Supplementary Fig. 6b). This result argues that it is actually IL-22-induced epithelial IL-18 that is upregulating Paneth cells for AIEC host defence. Taken together, our genetic evidence show that during AIEC infection, lamina propria-derived IL-22 and IL-22-induced epithelial IL-18 are functionally linked and coordinated to impact anti-microbial innate immunity of Paneth cells and subsequent adaptive IFNγ response in T cells.

**Differential regulation of intestinal organoids by IL-22 and IL-18**. As IL-22 injection promotes transit-amplifying (TA) compartments in vivo and loss of *Il-22* in mice compromises Ki67$^+$ proliferating TA cells (Fig. 2b-c)[15], we next asked whether the IL-22-IL-18 axis, or IL-18 itself, contributes to organoid growth, which is widely used to evaluate epithelial stemness in vitro[15–17]. As noted, recombinant IL-22 or IL-18 promoted Ki67$^+$ organoid proliferation in a Stat3-dependent manner (Fig. 6a). Ki67$^+$ crypts, more distributed in the TA compartments, were significantly reduced in *Il-22$^{-/-}$* and *Il-18$^{-/-}$* mice at the steady state or during AIEC infection (Figs. 2b–c, 6b-c). Therefore, IL-22 and IL-18 contribute to epithelial proliferation both in vitro and in vivo. While IL-22 injection into mice promotes epithelial proliferation by increasing crypt height or expanding TA compartments[15,17], new evidence suggest that IL-22 actually depletes Lgr5$^+$ stem cells via inhibiting Wnt/Notch signalling, which might adversely affect stem cell-mediated epithelial regeneration[17]. To investigate how IL-22 or IL-18 regulates epithelial stemness, we analyzed IL-22/IL-18-stimulated ileum organoids and found that IL-22 indeed inhibit stem cell genes (Ascl2, Olfm4) while IL-18 promotes stem cell genes (Lgr5, Ascl2, Olfm4), both in a Stat3-independent manner (Fig. 6d). Regarding morphological changes of organoids, we found that IL-22 shows a major effect to promote Stat3-mediated organoid size but IL-18 primarily promotes organoid budding in a Stat3-independent manner (Fig. 6e). While one study suggests that IL-22 promotes organoid size, but not

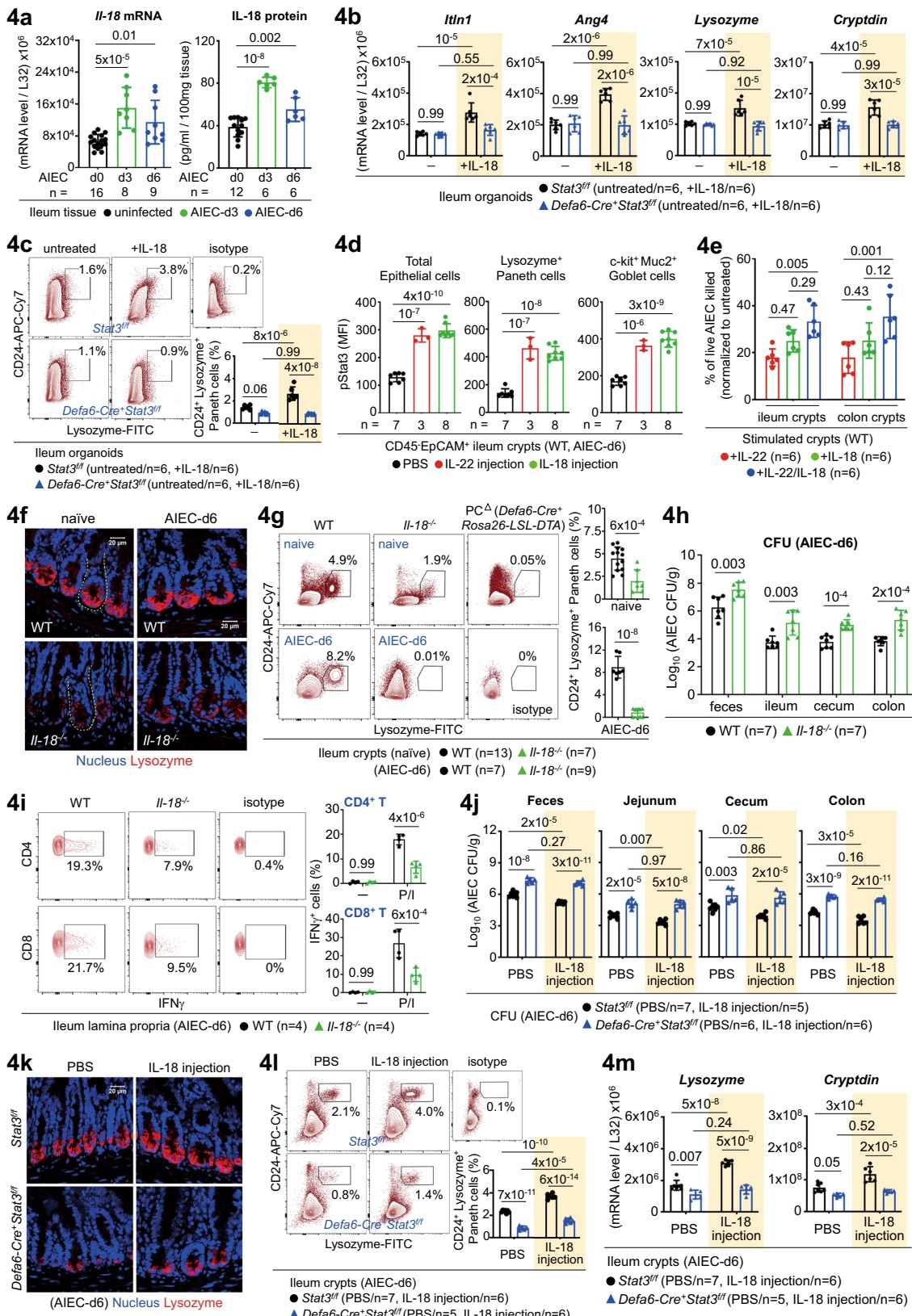

proliferation, by upregulating tight junction protein Claudin-2[17], we found that IL-22, but not IL-18, induces Claudin-2 in a Stat3-independent manner (Supplementary Fig. 7a). We further confirmed that IL-22 promotes organoid size independent of Paneth cells (or Stat3 in Paneth cells) and that IL-18 promotes organoid budding in a dose-dependent manner and the induction partially

depends on Paneth-cell Stat3 (Fig. 6f, Supplementary Fig. 7b). As Paneth cells provide crucial niche factors to support nearby stem cells and an organoid bud develops into a crypt-like structure[32], our results argue that IL-18-mediated organoid budding, compared to IL-22-mediated organoid size, is more associated with crypt formation or stem cell expansion. Nevertheless, while IL-22

**Fig. 4 IL-18 is a bona fide regulator of Paneth cells during AIEC host defence. a** Quantitative real-time PCR and ELISA analyses of ileum crypts or tissues for IL-18 in uninfected or AIEC-infected mice. **b** Quantitative real-time PCR analysis of IL-18-stimulated ileum organoids, derived from the indicated mice, for anti-microbial peptides. **c** Flow cytometry analysis of IL-18-stimulated ileum organoids, derived from the indicated mice, for CD24$^+$ Lysozyme$^+$ Paneth cells. **d** Flow cytometry analysis of ileum crypts, derived from IL-22 or IL-18-injected mice, for phospho-Stat3 (pStat3) in the indicated epithelial subsets. Mean fluorescence intensity (MFI) of pStat3 is indicated. **e** Fresh crypts were stimulated with IL-22, IL-18, or both, to induce the release of anti-microbial peptides (AMP). The AMP-containing supernatants were then incubated with live AIEC and the percentage of bacterial killing was determined by the CFU assay after normalized to unstimulated crypts. **f** Immunofluorescence analysis of ileum crypts (outlined with a dashed line) for Lysozyme$^+$ Paneth cells at the crypt base in the indicated mice. **g** Flow cytometry analysis of ileum crypts, derived from the indicated mice, for CD24$^+$ Lysozyme$^+$ Paneth cells. **h** Colony-forming unit (CFU) of AIEC was analyzed in feces and intestinal tissues isolated from the indicated mice at d6 post infection. **i** Flow cytometry analysis of PMA/Ionomycin (P/I)-stimulated ileum lamina propria cells for intracellular IFNγ in CD4$^+$ or CD8$^+$ T cells in AIEC-infected mice. **j** CFU of AIEC was analyzed in feces and intestinal tissues isolated from the indicated PBS or IL-18-injected mice at d6 post infection. **k** Immunofluorescence analysis of ileum crypts for Lysozyme$^+$ Paneth cells at the crypt base in the indicated IL-18-injected mice at d6 post AIEC infection. **l** Flow cytometry analysis of ileum crypts, derived from the indicated IL-18-injected mice, for CD24$^+$ Lysozyme$^+$ Paneth cells. **m** Quantitative real-time PCR analysis of ileum crypts from the indicated IL-18-injected mice, for Paneth cell marker Lysozyme and Cryptdin. Each symbol in bar graphs represents an ileum crypt sample (**a, d, e, g, l, m**), tissue sample (**a, h, i, j**), or organoid culture (**b, c**), derived from one mouse. Data shown are representative (**f, i, k**) or combined (**a–e, g, h, j, l, m**) results from two independent reproducible experiments. Figures 2f and 4c were performd in the same experiment. Statistical significance is indicated using unpaired two-tailed *t* test (**g, h**), One-way ANOVA with Sidak's multiple comparisons test (**a, d, e**), or Two-way ANOVA with Tukey's multiple comparisons test (**b, c, i, j, l, m**). Data are presented as mean ± SD. Source data are provided as a Source Data file.

inhibits stem cell genes (by *Ascl2/Olfm4* mRNA) slightly better in the absence of IL-18 (because IL-22-induced IL-18 promotes stem cells in the same culture), it still promotes organoid budding and size in *Il-18*$^{-/-}$ organoids (Fig. 6g-h). These discrepancies could be due to the limitation of organoid culture where many growth factors are added for optimal growth condition that may directly affect the function of IL-22 or IL-18. For example, recombinant R-spondin 1 is added to provide enhanced Wnt signalling for stemness[33] and Jagged-1 peptide (a Notch agonist) is used to enhance Notch signalling[34], both of which might potentially counteract IL-22-mediated suppression of Wnt/Notch signalling during organoid culture[17]. Collectively, while both IL-22 and IL-18 promote Stat3-dependent Ki67$^+$ organoid proliferation and size, as well as Stat3-independent organoid budding, IL-22 suppresses, but IL-18 promotes, mRNA levels of stem cell markers, both independently of Stat3. These results indicate that in vitro IL-22 and IL-18 mediate both redundant and non-redundant functions in proliferation, size, and budding of organoids, and that the cross-regulation of stem cells by IL-22 and IL-18 in vivo may not be faithfully recapitulated using in vitro organoid culture.

**IL-18-Akt-Tcf4 axis upregulates Lgr5$^+$ stem cells.** While our gain-of-function studies in IL-22 or IL-18-stimulated organoids support a role for IL-22 and IL-18 in epithelial proliferation (Fig. 6), loss-of-function study in genetic knockout mice is also crucial to elucidate the in vivo function of IL-22-Stat3 or IL-18 signalling in barrier homeostasis. To this end, we explored how loss of *Il-22*, *Il-18*, or epithelial *Stat3* in mice affects stem cells at the steady state or during AIEC infection. As noted, all knockout mice showed compromised Lgr5$^+$ stem cells in the context of homeostasis or AIEC host defence (Fig. 7a–c, Supplementary Fig. 8a-b), providing in vivo evidence that the IL-22-Stat3 axis or IL-18 signalling is a key regulator of stem cells. By immunofluorescence analysis of littermate mice, we also found that Olfm4$^+$ crypt base columnar (CBC) stem cells are greatly reduced in both *Il-22*$^{-/-}$ and *Il-18*$^{-/-}$ crypts at the steady state or during AIEC infection (Fig. 7d)[26,35]. Furthermore, correlated to mRNA data and organoid budding (Fig. 6d-e), we confirmed, by flow cytometry analysis, that IL-18 promotes Lgr5$^+$ stem cells independently of Stat3 in ileum organoids (Fig. 7e). Therefore, while Stat3 activation has been shown to contribute to Lgr5-β-catenin signalling for stemness and IL-18 indeed activate Stat3 in crypts (Fig. 4d)[36,37], our results exclude a role for IL-18-Stat3 signalling in Lgr5$^+$ stem cell upregulation. To further elucidate the underlying mechanism, we

asked how IL-18 is involved in Wnt-mediated signalling for stemness, which is induced via the Lgr5-Frizzled-Lrp5/6 complex leading to β-catenin/Tcf4 activation[32]. T-cell factor 4 (Tcf4) is a key transcription factor downstream of Wnt signalling and is indispensable for epithelial stem cell compartments in vivo[38,39]. We analyzed the mouse *Lgr5* promoter and identified six (marked as P1~P6) putative TCF4 consensus binding sites (Supplementary Fig. 8c). Further analysis of IL-22 or IL-18-stimulated ileum organoids by the ChIP assay, with a specific anti-Tcf4 antibody for immunoprecipitation, revealed that IL-18, but not IL-22, specifically promotes Tcf4 binding to the P1 region within the *Lgr5* promoter (Fig. 7f). Next, as Akt activation has been linked to the survival of Lgr5$^+$ organoids[40], we tested in colon epithelial CMT93 cells by Western blotting and found that IL-18 robustly induces Akt phosphorylation at Ser$^{473}$ and a corresponding induction of Lgr5. Inhibition of Akt activity with an inhibitor MK-2206 abolished IL-18-induced Akt activation and reduced Lgr5 expression as well (Fig. 7g)[41], indicating that the IL-18-Akt-Lgr5 pathway is acting on stemness induction. To further confirm this, a ChIP assay in IL-18/Akt inhibitor-treated CMT93 cells was performed. The result showed that the Akt blockade indeed abolish the IL-18-induced binding of Tcf4 to the P1 region within the *Lgr5* promoter (Fig. 7h), supporting that it is the IL-18-Akt-Tcf4-Lgr5 axis that is crucial for stem cell upregulation. Taken together, in an unique and non-redundant manner compared to IL-22-mediated cascade, the IL-22-initiated IL-18 circuit promotes Stat3-independent but Tcf4-dependent transcriptional induction of Lgr5 via Akt activation. Furthermore, while being directly induced by the IL-22-Stat3 axis, epithelium-derived IL-18 has an autocrine function in epithelial cells, which is stimulatory, but not inhibitory in the case of IL-22, towards Lgr5$^+$ stem cells.

**IL-22 promotes mucin secretion via IL-18 during AIEC infection.** As IL-22 and IL-18 are involved in Goblet cell-mediated mucosal infection including helminth and Salmonella[42,43], we next asked how IL-22 or IL-18 regulates Goblet cells during AIEC infection and tested whether these two cytokines are crosstalked. First, we found that the percentage (%) and level (MFI) of IL-18R expression are higher than those of IL-22R in Muc2$^+$ Goblet cells and loss of *Il-22* or *Il-18* in mice does not alter the expression patterns (Supplementary Fig. 9a). However, the percentage of Muc2$^+$ Goblet cells in both knockout mice was reduced at the steady state, as was also observed in *Muc2* mRNA levels in ileum epithelial cells isolated from naïve or AIEC-infected mice (Fig. 8a, Supplementary Fig. 9b). Correlated to mRNA levels, by

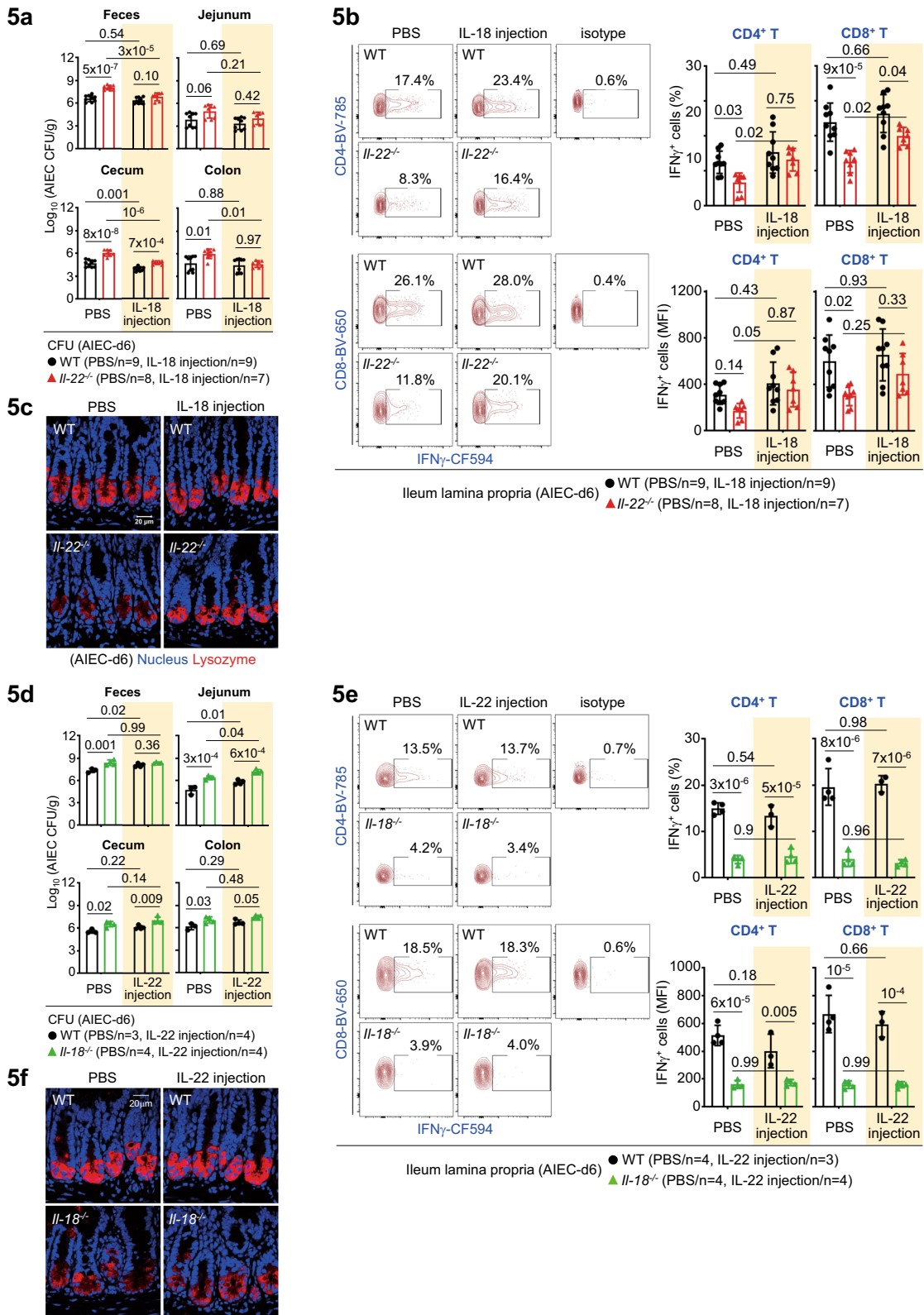

immunofluorescence and H&E staining, we confirmed that loss of *Il-22* or *Il-18* in mice causes a homeostatic reduction of Muc2+ Goblet cells (Fig. 8b and Supplementary Fig. 9c, upper panels). As noted, the reduction at the steady state was not due to aberrant apoptosis of Goblet cells by TUNEL staining (Supplementary Fig. 9d, upper panels). Intriguingly, AIEC infection promoted drastic

mucin secretion in wild-type mice but this process was almost abolished in *Il-22*−/− or *Il-18*−/− mice (Fig. 8b and Supplementary Fig. 9c, lower panels). Further analysis of IL-18-injected *Il-22*−/− mice confirmed that IL-18 indeed promote mucin secretion from Goblet cells during AIEC infection (Fig. 8c). In contrast, IL-22 injection into *Il-18*−/− mice failed to promote mucin secretion,

**Fig. 5 IL-22 links IL-18 for IFNγ and Paneth cell responses during AIEC infection. a** Colony-forming unit (CFU) of AIEC was analyzed in feces and intestinal tissues isolated from the indicated PBS or IL-18-injected mice at day-6 (d6) post infection. **b** Flow cytometry analysis of PMA/Ionomycin-stimulated ileum lamina propria cells for intracellular IFNγ in CD4+ or CD8+ T cells in the indicated PBS or IL-18-injected mice. **c** Immunofluorescence analysis of ileum crypts for Lysozyme+ Paneth cells at the crypt base in the indicated mice at d6 post infection. **d** CFU of AIEC was analyzed in feces and intestinal tissues isolated from the indicated PBS or IL-22-injected mice at d6 post infection. **e** Flow cytometry analysis of PMA/Ionomycin-stimulated ileum lamina propria cells for intracellular IFNγ in CD4+ or CD8+ T cells in the indicated PBS or IL-22-injected mice. **f** Immunofluorescence analysis of ileum crypts for Lysozyme+ Paneth cells at the crypt base in the indicated mice at d6 post infection. Each symbol in bar graphs represents an intestinal tissue sample (**a**, **b**, **d**, **e**) isolated from one mouse. Data shown are representative (**c-f**) or combined (**a**, **b**) results from two independent reproducible experiments. Statistical significance is indicated using Two-way ANOVA with Tukey's multiple comparisons test (**a**, **b**, **d**, **e**). Data are presented as mean ± SD. Source data are provided as a Source Data file.

which is reminiscent of the results that IL-22 injection also failed to restore Paneth cell or IFNγ response in $Il$-$18^{-/-}$ mice during AIEC infection (Fig. 5e-f). Again, our results argue that it is actually IL-22-induced epithelial IL-18 that is upregulating Goblet cells for AIEC host defence. Furthermore, in ileum organoids, it appears that both IL-22 and IL-18 promote $Muc2$ mRNA levels in a Stat3-dependent manner and that IL-22 induces $Muc2$ mRNA independent of epithelial IL-18 (Fig. 8d, Supplementary Fig. 9e). Of note, while there was increased cell apoptosis by TUNEL staining in both $Il$-$22^{-/-}$ and $Il$-$18^{-/-}$ mice during AIEC infection, the percentage of Caspase-3+ UEA1+ Goblet cells was not increased in $Il$-$18^{-/-}$ mice (Supplementary Fig. 9d, f). Overall, these results indicate that IL-18 regulates mucin production and secretion but not survival of Goblet cells. Taken together, integrated and differential regulation of epithelial barrier by IL-22 and IL-18, mostly based on organoid culture, is summarized (Fig. 8e). We conclude that mucosal AIEC infection triggers an early IL-22 response cascade which subsequently initiates an IL-18 response circuit. This IL-18-mediated circuit promotes IL-22-redundant and non-redundant immunity by boosting proliferating stem/TA cells, anti-microbial Paneth cells, mucin-producing Goblet cells, and IFNγ-producing T cells.

## Discussion

Intestinal microbiota dysbiosis is an important factor in the etiology of Crohn's disease (CD) and high prevalence of adherent-invasive $Escherichia coli$ (AIEC) in the mucosa of CD patients has been consistently reported[44,45]. Both IL-22 and IL-18 contribute to active CD and represent a promising therapeutic potential[9,19,21,29]. Here, our work provides clinical relevance of these two cytokines to host defence against Crohn's AIEC, by revealing a crucial role of the IL-22-IL-18 axis and a coordinated IL-22-initiated IL-18 response circuit at the frontline barrier.

The IL-22 response cascade and IL-18 response circuit exert redundant and non-redundant immunity for host defence. At the cellular level, we show that IL-18 regulates Ki67+ proliferating transit-amplifying (TA) cells, Lgr5+ stem cells, and anti-microbial Paneth cells at the steady state and during AIEC host defence, providing the first genetic evidence that IL-18 is a bona fide regulator of stem cells and Paneth cells. At the molecular level, we reveal that IL-22 directs a pleiotropic IL-18 response circuit, which subsequently deploys an IL-18-Stat3-AMP (targeting Paneth cells) pathway, an IL-18-Akt-Tcf4-Lgr5 (targeting stem cells) pathway, and an IL-18-IFNγ (targeting CD8+ T cells) pathway, to boost innate immunity and connect innate to adaptive immune responses. Our rescue experiments in mice further provide in vivo evidence, showing a direct link of the IL-22-IL-18-IFNγ axis and the IL-22-IL-18-Paneth cell axis to AIEC host defence. As IL-22/IL-18-mediated therapeutic strategy is under development for treating inflammatory diseases and cancers[29], our work here provides new insights of how the interplay of IL-22 and IL-18 contributes to cross-regulated immune response.

Increased IL-18 levels in serum and mucosal biopsies are correlated to Crohn's disease and IL-18 upregulation is reported to be a feature of CD[19,20]. As a potent IFNγ inducer in T cells, IL-18 could be an ideal target for blocking inflammatory Th1-mediated IFNγ response in CD patients[19]. Intriguingly, mucosal biopsies of IBD patients show more abundant IL-18 expression in epithelial cells over lamina propria mononuclear cells[46], suggesting that the function of IL-18 in epithelial cells might be underestimated, especially in wound healing and anti-microbial immunity which are key processes for tissue recovery and host defence. In this regard, we reveal that IL-18 activates Akt-Tcf4 to promote Lgr5+ stem cell expansion or tissue regeneration, which is highly relevant to CD pathogenesis and Lgr5-mediated intestinal tumorigenesis[47]. As a transcription factor, Tcf4 is a key regulator downstream of Wnt/β-catenin signalling[39]. In the gut, Tcf4 controls mucosal homeostasis and immunity as it directly binds to and activates the promoters of stem cell marker Lgr5 and Paneth-cell marker α-defensin[39,48]. Global deletion of $Tcf4$ in mice causes absence or abnormality of proliferating crypt compartments leading to neonatal lethality within 24 hr[38], phenotypically similar to global knockout of $Lgr5$[49]. Conditional $Tcf4$ ablation in gut epithelium causes loss of proliferating crypt cells, as well as gradual depletion of stem cell compartment and nearby Paneth cells[50]. These facts explain why Tcf4 is a pronounced risk factor for Crohn's disease[51], as is also evidenced by a high correlation of genetic variants of $Tcf4$ and the decrease of Paneth cell-specific α-defensin HD5 to CD patients[52,53]. Therefore, the IL-18-Akt-Tcf4-Lgr5 pathway may represent a novel and key repair mechanism in the epithelium to prevent CD pathogenesis. In addition, while activation of the IL-18-IFNγ axis in T cells appears detrimental to the mucosa, we found the IL-18-mediated IFNγ response in CD8+ T cells is also essential for clearance of AIEC[5], which is highly associated with ileum mucosa in CD patients[3]. Together, our results support a protective role for IL-18 in host defence, where IL-18 is promoting IFNγ+ T cells with IL-12, stem cells via Tcf4, and Paneth cells via Stat3. As such, concerns should be taken when the IL-18 blockade, which can suppress epithelial barrier function and T-cell-mediated host defence, is developed for treating mucosal inflammation such as Crohn's disease.

Our study also provides a comprehensive dissection of the controversial role for IL-22 and the unexplored function for IL-18 in stem cells[15,17,54]. While accumulating evidence show that IL-22 promotes proliferative transit-amplifying (TA) compartments but suppresses Lgr5+ stem cells in vivo and in vitro[15–17], there have been no reports of a role for IL-18 in stem cells. In organoid culture, we found that both IL-22 and IL-18 promote Stat3-independent budding and Stat3-dependent size expansion. As an organoid bud develops into a crypt-like structure that usually contains 4~6 stem cells[32,55], we reason that the number of organoid buds is a good indicator of stemness. Intriguingly, IL-18-induced organoid budding and increase in size are correlated to IL-18-induced mRNA upregulation of stem cell markers (Lgr5,

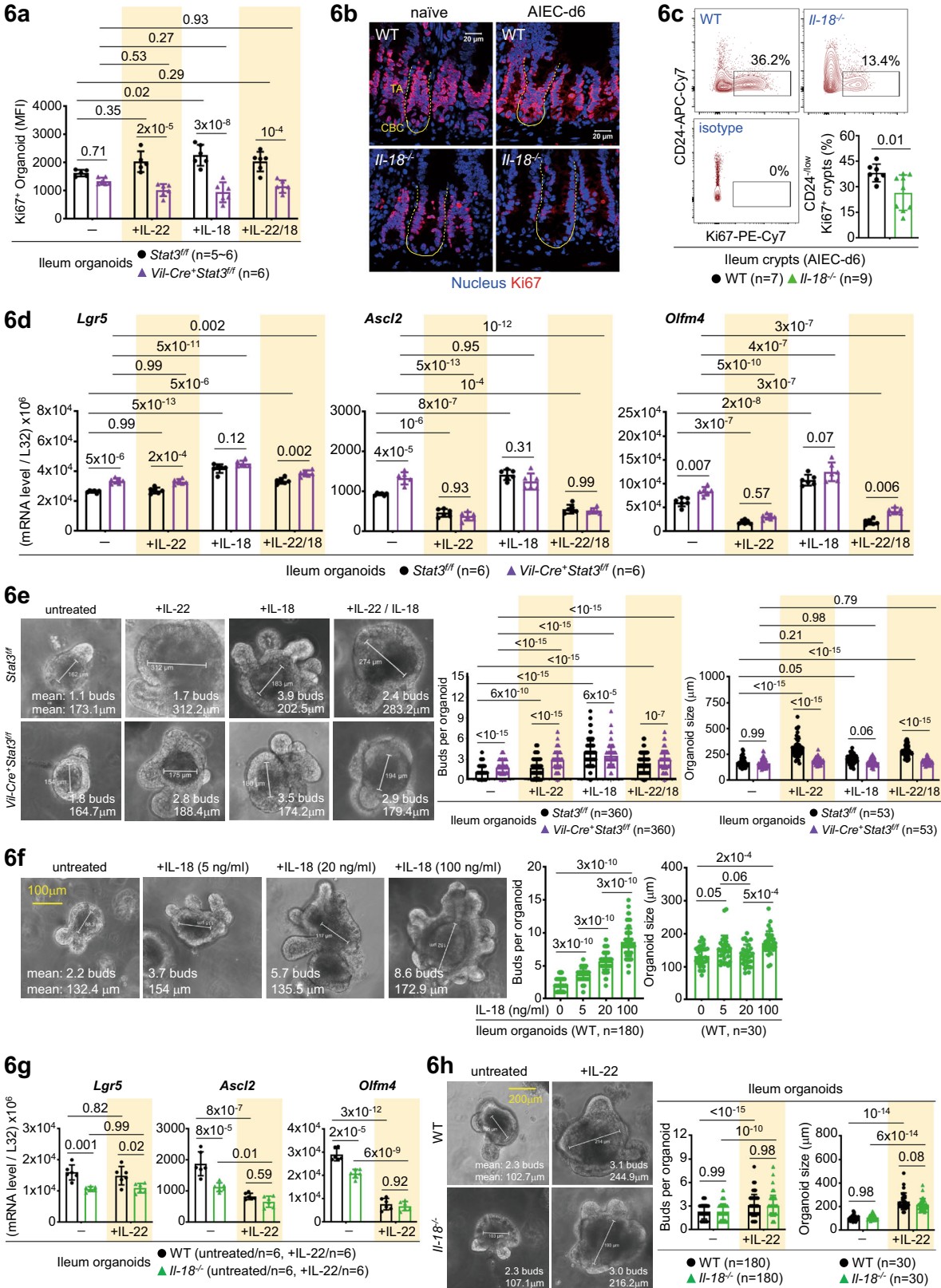

Ascl2, Olfm4), but similar IL-22-induced morphological changes can not be attributed to IL-22-induced mRNA suppression of stem cell markers. This discrepancy immediately argues that organoids may not be a truly untouched sample after cultured in an optimized condition, which usually provides enhanced Wnt/Notch signalling for growth that potentially may interfere with

IL-22-mediated signalling[17]. However, this also explains why IL-22-mediated organoid budding and size expansion does not require IL-18, which is supported by our identification of non-redundant IL-18-Akt-Tcf4-Lgr5 pathway for stemness. In contrast, by flow cytometry and immunofluorescence analyses of fresh "untouched" crypt samples for stem cells and TA cells, we

**Fig. 6 Differential role of IL-22 and IL-18 in organoid culture. a** Flow cytometry analysis of IL-22 or IL-18-stimulated ileum organoids, derived from the indicated mice, for Ki67+ proliferating cells. Mean fluorescence intensity (MFI) of Ki67+ organoids is shown. **b** Immunofluorescence analysis of ileum crypts for Ki67+ proliferating cells in the indicated naïve or AIEC-infected mice. Crypt base columnar (CBC, outlined with a solid line) stem cell and the above transit-amplifying (TA, outlined with a dashed line) compartments are indicated. **c** Flow cytometry analysis of ileum crypts for CD24-/low Ki67+ proliferating cells in the indicated AIEC-infected mice. **d** Quantitative real-time PCR analysis of IL-22 or IL-18-stimulated ileum organoids, derived from the indicated mice, for stem cell marker Lgr5, Ascl2, and Olfm4. **e** Quantification of the number of buds per organoid and size in IL-22 or IL-18-stimulated ileum organoids, derived from the indicated mice. "n" indicates the number of images taken from organoids derived from four mice in each group. **f** Quantification of the number of buds in IL-18-stimulated wild-type ileum organoids. "n" indicates the number of images taken from organoids derived from three mice per group. **g** Quantitative real-time PCR analysis of IL-22-stimulated ileum organoids, derived from the indicated mice, for stem cell marker Lgr5, Ascl2, and Olfm4. **h** Quantification of the number of buds per organoid and size in IL-22-stimulated ileum organoids, derived from the indicated mice. "n" indicates the number of images taken from organoids derived from four mice per group. Each symbol in bar graphs represents an ileum crypt sample (**c**) or organoid culture (**a**, **d**, **g**), derived from one mouse. Data shown are representative (**b**) or combined (**a**, **c–h**) results from two independent reproducible experiments. Statistical significance is indicated using unpaired two-tailed *t* test (**c**), One-way ANOVA with Sidak's multiple comparisons test (**f**), or Two-way ANOVA with Tukey's multiple comparisons test (**a**, **d**, **e**, **g**, **h**). Data are presented as mean ± SD. Source data are provided as a Source Data file.

found that loss of *Il-22*, epithelial *Stat3*, or *Il-18* in mice indeed cause a reduction of Lgr5+ and Ki67+ crypts at the steady state and during AIEC infection. Together, these results support that IL-22-Stat3 and IL-18 signalling are indeed crucial for the development of stem cell and TA compartments in vivo.

There are few reports to study the role of IL-22 in Paneth cells and the function of IL-18 in Paneth cells has not been revealed. Recent studies show that IL-22 promotes Lysozyme+ cells in organoids or in mice after IL-22 injection[16,17]. While an earlier study shows IL-22R is not detectable on Lysozyme+ Paneth cells and therefore has no effect on Paneth cell frequency in organoids[15], new genetic evidence in mice indicate that IL-22R signalling in Paneth cells is crucial for their maturation, which consequently might affect microbiota colonization and anti-bacterial immunity[18]. We extend these observations by showing that, first, both IL-22 and IL-18 indeed induce Stat3 activation in Paneth cells and both require Stat3 to promote Lysozyme+ Paneth cells. Secondly, IL-18R is highly expressed in CD24+ Paneth cells but IL-22R is more enriched in CD24-/low subsets which mostly contain stem cells and TA cells, suggesting that IL-22 and IL-18, while functionally linked, may preferentially target distinct epithelial subsets. Thirdly, different from the action mode in stem cells, IL-22 absolutely requires IL-18 to promote Paneth cells in vitro and in vivo. Lastly, in sorted Paneth cells, we show that IL-18 is able to directly promote AMP production from Paneth cells. Therefore, the identification of the IL-22-IL-18-Stat3 axis in Paneth cell functionality could impact the therapeutic strategy of targeting Paneth cell-mediated host defence in Crohn's disease (also called Paneth disease)[56].

In addition to differential expression of the receptor, the requirement for signalling components downstream of IL-22R and IL-18R could also be different in epithelial subsets. IL-22R-associated Jak1/Tyk2 activates Stat3 when IL-22 signalling is triggered[57]. It is unclear whether the same machinery is involved in IL-18-mediated Stat3 activation in Paneth cells but it is Akt that relays IL-18 signalling to activate Tcf4-Lgr5 in stem cells. Canonically, the IL-18 receptor complex interacts with Myd88, which is capable of inducing downstream NF-κB via IRAK1-4, or MAPK via TRAF6[10]. Therefore, the differential receptor expression pattern and requirement for different signalling components in IL-22 vs IL-18 signalling might highlight the importance and uniqueness of IL-18 response circuit, downstream of the IL-22 response cascade, in the homeostasis of epithelial barrier and during AIEC host defence.

## Methods

**Animals**. *Il-18−/−* C57BL/6 mice (stock #4130), *Ifnγ −/−* C57BL/6 mice (stock #2287), *Vil-Cre* C57BL/6 mice (stock #4586), *Stat3flox/flox* (*Stat3f/f*) C57BL/6 mice (stock #16923), and *Rosa26-LSL-DTA* C57BL/6 mice (stock #9669) were obtained

from The Jackson Laboratory. *Il-22−/−* C57BL/6 mice were kindly provided by Dr. Wenjun Ouyang[58], Department of Immunology, Genentech. *Defa6-Cre* mice were kindly provided by Dr. Richard S. Blumberg[59], Department of Medicine, Brigham and Women's Hospital, Harvard Medical School. *Stat3f/f* mice were bred with *Defa6-Cre* mice (for Paneth cell-specific) or *Vil-Cre* mice (for epithelium-specific), to generate conditional knockout mice. *Defa6-Cre* mice were bred with *Rosa26-LSL-DTA* mice to generate Paneth cell-deficient (PCΔ) mice. 8-12 week-old, age/gender-matched littermate mice (littermate WT vs *Il-22−/−* or *Il-18−/−*, littermate *Stat3f/f* vs *Vil-Cre+Stat3f/f* or *Defa6-Cre+Stat3f/f*) were used in all experiments. Wild-type controls were generated by intercrossing of heterozygous knockout mice. Animals were bred separately, not co-housed, and maintained in a specific-pathogen-free (SPF) facility at a relative humidity 50 ± 10%, 20–26 °C, and in 12 h dark/light cycles (08:00–20:00 light). Experiments were performed on mature animals (8-12 week-old) in both male and female mice, unless otherwise indicated. Animal care and experimental protocols (Protocol ID: 17-05-1092) have been approved by the Institutional Animal Care and Use Committee (IACUC) at the Institute of Biomedical Sciences, Academia Sinica. Ethical compliance has been observed in all animal studies. Dr. John T. Kung is the chairperson of IACUC and Ethics Committee in Academia Sinica, Taiwan.

**AIEC infection and in vivo rescue experiment**. Mice (8-12 week-old) were orally given 40 mg of ampicillin in a total volume of 200 μl one day before oral gavage with 2×10⁹ colony forming units (CFU) of adherent-invasive *E. coli* strain (AIEC) NRG875c (O83:H1) (a gift of Dr. Brian K. Coombes[5]) in a total volume of 200 μl per mouse. The bacteria were prepared by sharking at 37 °C overnight in LB broth. The concentration of bacteria was measured by serially diluted and plated each inoculation culture to confirm the colony-forming units (CFU) administrated. Mice were euthanized at day-3 or day-6 for desired experiments, when they usually show no detectable pathology by H&E staining which is consistent with the original report[5]. Some mild alterations of Paneth cell or Goblet cell numbers can be microscopically observed during AIEC infection. Feces were collected into 1 ml cold PBS buffer and homogenized by a vortexer. The ileum, cecum, and colon were collected into a 50 ml tube containing 10 ml cold PBS buffer and homogenized using the tissue homogenizer (OMNI, #TH115) with plastic homogenizing probes (OMNI, #30750H-S). Feces and tissue homogenates were serially diluted and plated on LB agar containing ampicillin to select for AIEC NRG875c. After 24 h of incubation at 37 °C, colony-forming units (CFU) were counted and normalized to the weight of fecal samples as indicated. For administration of IL-22 and IL-18 in vivo, groups of mice were intraperitoneally injected with recombinant mouse IL-22 (Biolegend #576206) or IL-18 (Biolegend #767004) for 5 continuous days (day-1~day-5) after AIEC infection (day-0) at the dose of 1 μg in 400 μl of PBS per mouse. The control group received PBS only.

**Antibodies**. The following primary antibodies were used for Western blots: rabbit anti-cleaved Caspase-3 (Asp175) [5A1E] (1:1000, Cell Signaling Technology #9664), rabbit anti-Lgr5 [EPR3065Y] (1:1000, Abcam #ab75850), rabbit anit-IL-18 (1:1000, Proteintech #10663-1-AP), rabbit anti-phospho-Stat3Tyr705 [D3A7] (1:1000, Cell Signaling Technology #9145), mouse anti-Stat3 [124H6] (1:1000, Cell Signaling Technology #9139), mouse anti-β-Actin [C4] (1:1000, Santa Cruz Biotechnology #sc-47778), rabbit anti-α-Tubulin (1:1000, Cell Signaling Technology #2144), rabbit anti-Akt [N3C2] (1:2000, GeneTex #GTX121937), rabbit anti-phospho-Akt (Ser473) [D9E] (1:1000, Cell Signaling Technology #4060), rabbit anti-Noggin [FL-232] (1:1000, Santa Cruz #sc-25656), and rabbit anti-R-spondin [C13]-R (1:1000, Santa Cruz #sc-49090). HRP-conjugated secondary antibodies (1:10000 dilution) for immunoblotting were from Jackson ImmunoResearch unless otherwise mentioned. The following secondary antibodies were used: goat anti-mouse IgG (#115-035-003), goat anti-rabbit IgG (#111-035-003). Western blots were scanned with ImageQuant LAS 4000 mini equipment and its software. The intensity of each band was quantified by ImageJ (1.53k) software. The following

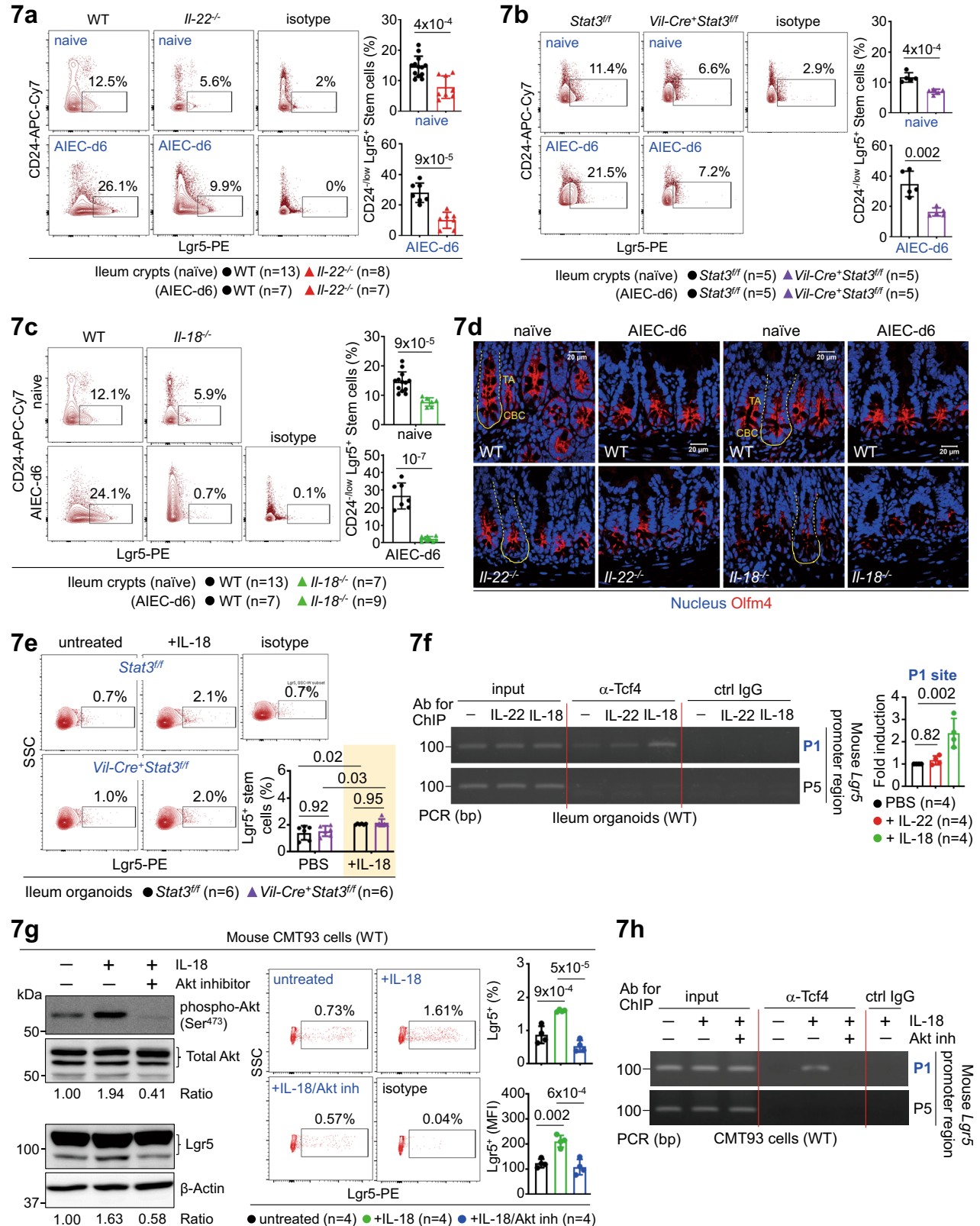

primary antibodies were used for ChIP: rabbit anti-phospho-Stat3[Tyr705] [D3A7] (Cell Signaling Technology #9145), rabbit anti-TCF4 /TCF7L2 [C48H11] (Cell Signaling Technology #2569). The following primary antibodies were used for immunofluorescence: rabbit anti-Ki67 [SP6] (1:200, Abcam #ab16667), mouse anti-Lysozyme (1:200, St John's Laboratory #STJ16101308), rabbit anti-Lysozyme [EPR2994(2)] (1:250, Abcam #ab108508), rabbit anti-Olfm4 [D6Y5A] (1:200, Cell Signaling Technology #39141), rabbit anti-Muc2 [H-300] (1:100, Santa Cruz #sc-15334), mouse anti-E-Cadherin [4A2] (1:200, Cell Signaling Technology #14472).

Fluorophore-conjugated secondary antibodies (all 1:300 dilution used) for immunofluorescence were purchased from Cell signaling Technology unless otherwise mentioned. The following secondary antibodies were used: anti-mouse Alexa Fluor-594 (#8890), anti-rabbit Alexa Fluor-488 (#4412), anti-rabbit Alexa Fluor-594 (#8889), anti-rat Alexa Fluor-488 (#4416), and anti-rat Alexa Fluor-594 (Invitrogen #A-11007). Confocal images were obtained with a Carl Zeiss LSM 700 stage imaging system under a 20x or 40x oil-immersion objectives with ZEN 2011 software. The following primary antibodies were used for flow cytometry:

**Fig. 7 IL-18-Akt-Tcf4 signalling upregulates Lgr5$^+$ stem cells. a–c** Flow cytometry analysis of CD24$^{-/low}$ ileum crypts, isolated from the indicated uninfected or AIEC-infected mice at d6, for Lgr5$^+$ stem cells. An isotype antibody was included to indicate the specificity of the Lgr5 antibody. **d** Immunofluorescence analysis of ileum crypts for Olfm4$^+$ stem cells in the indicated uninfected or AIEC-infected mice at d6. Crypt base columnar (CBC, outlined with a solid line) stem cell and the above transit-amplifying (TA, outlined with a dashed line) compartments are indicated. **e** Flow cytometry analysis of IL-18-stimulated ileum organoids, derived from the indicated mice, for Lgr5$^+$ stem cells. **f** Chromatin immunoprecipitation (ChIP) and PCR analyses of IL-22 or IL-18-stimulated wild-type ileum organoids for the binding of Tcf4 to the mouse *Lgr5* promoter regions (P1-P6). Fold induction is calculated based on PCR signal strength of Tcf4-bound *Lgr5* promoter P1 region before and after IL-22 or IL-18 stimulation. **g** Western blot and flow cytometry analyses of IL-18/Akt inhibitor-stimulated CMT93 cells for Akt phosphorylation and Lgr5. Quantification and the ratio (pAkt to total Akt, Lgr5 to β-actin) of protein bands are indicated. **h** ChIP and PCR analyses of IL-18/Akt inhibitor-stimulated CMT93 cells for the binding of Tcf4 to the mouse *Lgr5* promoter regions (P1 and P5). Fold induction is calculated based on PCR signal strength of Tcf4-bound *Lgr5* promoter P1 region before and after stimulation. Each symbol in bar graphs represents a well of CMT93 cell culture (**g**), an ileum crypt sample (**a–c**) or organoid culture (**e, f**) derived from one mouse. Data shown are representative (**d, f–h**) or combined (**a–c, e**) results from two independent reproducible experiments. Statistical significance is indicated using unpaired two-tailed *t* test (**a–c**), One-way ANOVA with Sidak's multiple comparisons test (**f, g**), or Two-way ANOVA with Tukey's multiple comparisons test (**e**). Data are presented as mean ± SD. Source data are provided as a Source Data file.

fixable viability dye eFluor 780 (1:2000, eBioscience #65-0865-18), rat anti-CD326 (EpCAM) [G8.8] BV510 (1:400, BD #747748), rat anti-CD326 (EpCAM) [G8.8] BV421 (1:200, BD #563214), hamster anti-CD3e [145-2C11] FITC (1:200, eBioscience #11-0031-85), rat anti-CD3 [17A2] BV786 (1:200, BD # 564010), rat anti-CD45 [30-F11] Alexa Fluor 700 (1:400, eBioscience #56-0451-82), rat anti-CD45 [30-F11] APC (1:400, Biolegend #103112), rat anti-CD4 [RM4-5] Brilliant Violet 785 (1:200, Biolegend #100552), rat anti-CD4 [RM4-5] eFluor 450 (1:200, eBioscience #48-0042-82), rat anti-CD8a [53-6.7] Brilliant Violet 650 (1:200, Biolegend #100742), rat anti-CD11b [M1/70] PE (1:200, Biolegend #101208), rat anti-Ly-6G/Ly-6C (Gr1) [RB6-8C5] PE-Cyanine7 (1:400, eBioscience #25-5931-82), mouse anti-NK1.1 [PK136] Alexa Fluor 700 (1:200, eBioscience #56-5941-82), rat anti-CD90.2 (Thy1.2) [30-H12] PerCP-eFluor 710 (1:400, eBioscience #46-0903-82), rat anti-CD90.2 (Thy1.2) [53-2.1] V500 (1:400, BD #561616), rat anti-CD24 [M1/69] APC-eFluor 780 (1:200, eBioscience #47-0242-82), rat anti-CD24 [M1/69] BV605 (1:400, BD #563060), rabbit anti-Lysozyme FITC (1:200, Dako #F0372), rat anti-c-kit [ACK2] PE/Dazzle 594 (1:200, Biolegend #135128), rat anti-c-kit [2B8] APC/Cyanine7 (1:200, Biolegend #105826), mouse anti-CD66a [CC1] APC (1:200, eBioscience #17-0661-80), rat anti-IFNγ [XMG1.2] PE-CF594 (1:400, BD #562303), mouse anti-IL-18 [12E7.1] unconjugated (1:200, Sigma #04-1585), rat anti-Ki-67 [SolA15] PE-Cyanine7 (1:400, eBioscience #25-5698-82), rat anti-Lgr5/GPR49 [#803420] PE (1:400, R&D #FAB8240P), rabbit anti-active Caspase-3 [C92-605] Alexa Fluor-647 (5 µl /test, BD Biosciences #560626), rat anti-IL-22Ra1 [#496514] APC (1:20, R&D #FAB42941A), Mouse anti-IL-18R1 [P3TUNYA] PE (1:200, eBioscience #12-5183-82). The secondary antibody for IL-18 is goat anti-mouse IgG Alexa Fluor 647 (1:200, Invitrogen #A-21235), and the secondary antibody for Muc2 is goat anti-rabbit Alexa Fluor 488 (1:200, Cell Signaling Technology #4412). Flow cytometry data were collected by BD LSRII with BD FACS Diva software (v6.1.3) and analyzed by FlowJo (v10.0.7). Sorted cells were collected by BD FACSAria IIIu with BD FACS Diva software (v6.2).

**Tissue cytokine analysis by ELISA**. After cleaning and cut open longitudinally, ileum fragments were further cut into 1-2 mm pieces, transferred to a 24-well plate with 100 mg ileum/1 ml/well in RPMI buffer containing 10% fetal bovine serum (FBS) (Gibco #10437-028), 100 U/ml of penicillin, 100 µg/ml of streptomycin (Gibco #15140-122), and 10 µl/ml Gentamicin (Gibco #15750-078), and incubated in a 37 °C humidified 5% CO$_2$ incubator. After overnight (16 hr) incubation, supernatants were harvested and analyzed for protein levels by mouse IL-22 ELISA kit (ABclonal #RK00108) or mouse IL-18 ELISA kit (ABclonal #RK00104) according to instructions.

**Isolation, culture, and analysis of ileal lamina propria (LP) cells**. To isolate intestinal lamina propria cells, the described protocol was performed with some modifications[60]. Mice (8-12 week-old) were euthanized with CO$_2$ and the abdomen was cut open to separate ~12 cm of distal small intestine (whole Ileum). Peyer's Patches and fat tissues were removed and ileum fragments were cut open longitudinally and washed with PBS. The fragments were further cut into 2 cm pieces and then placed into 50 ml falcon tubes and washed vigorously by shaking in Hank's Balanced Salt Solution (HBSS) until the supernatant was clear. Ileum pieces were incubated with 40 ml of warm depletion buffer (1 mM EDTA in 1X PBS) in 100 ml glass bottles and stirred at 500 g at 37 °C for 20 min, then washed in 1X PBS by hand shaking vigorously in a 50 ml tube to ensure removal of epithelial cells. Next, ileum pieces were minced in 1 mm by a scissor in an eppendorf with 1 ml of digestion buffer [2% FBS, 15 µg/ml Liberase (Roche, #05401119001), 50 µg/ml DNase I (Sigma, #DN25) in RPMI medium (Gibco, #31800022)], then transferred to a 50 ml flask with 9 ml of digestion buffer and put for stirring at 500 rpm at 37 °C for 30 min. The supernatant was filtered through a 70 µm strainer (BD Biosciences #352350) into a new 50 ml tube on ice and the rest of pieces were homogenized by a 18 G needle then again stirred in 10 ml of digestion buffer for another 30 min. The supernatant was centrifuged at 600 g for 6 min at 4 °C and the

pelleted LP cells were resuspended in complete RPMI medium (10% fetal bovine serum, 100 U/ml of penicillin and 100 µg/ml of streptomycin in RPMI medium) for cell counting. The resulting cells were split into three aliquots for RNA extraction, unstimulated, or stimulated culture in a V-bottom 96-well plate for flow cytometry analysis of cell composition or cytokine production. After 4 hr stimulation of IL-23 (50 ng/ml, eBioscience, #14-8231), PMA (50 ng/ml, sigma, #P8139), and Ionomycin (800 ng/ml, sigma, #I9657), the cells were briefly resuspended by pipetting before spinning down at 1,500 rpm for 5 min at 4 °C and then washed once with staining buffer (2% FBS in 1X cold PBS). 50 µl staining buffer containing blocking antibody (anti-CD16/32) was added to cells for 10 min before performing surface staining for 20 min in dark at 4 °C. For intracellular staining, the cells were fixed and permeabilized in BD Fixation/Permeabilization Solution (BD, #554714) for 30 min at 4 °C, washed once with 200 µl of BD Perm Wash Buffer (BD, #554714), and followed by centrifugation at 1,800 rpm, 6 min at 4 °C. The cells were resuspended with intracellular staining antibody in 1X Wash Buffer and stained for 30 min in dark at 4 °C. After washing in staining buffer, the cells were resuspended in 200 µl staining buffer and immediately analyzed by BD LSR-II flow cytometer.

**Intestinal crypts isolation for oganoid culture**. To isolate the crypts, the described protocol was performed with some modificatins[61]. The clean ileum fragments were cut into 2 cm pieces and then placed into 50 ml falcon tubes and washed vigorously by shaking in Hank's Balanced Salt Solution (HBSS) until the supernatant was clear. To dissociate the crypts from ileum, fragments were placed in a 50 ml tube containing 10 ml of 30 mM EDTA in PBS buffer and shaked at 100 rpm in 37 °C for 5 min. The PBS-EDTA buffer was aspirated, replaced with cold wash buffer [1X PBS pH 7.4, 1X penicillin/streptomycin, 50 µg/ml Gentamicin, 0.1% bovine serum albumin (BSA)], and sharked vigorously with a vortexer 4 times (2 s at a time) to remove villi and epithelial debris. After that, the supernatant was aspirated and 10 ml cold wash buffer was added to the fragments and vortexed for 8 times (2 s at a time). The supernatant containing crypts was collected and transferred to a new 50 ml falcon tube pre-coated with 5% FBS. This step was repeated and each successive fraction was collected and an aliquot was examined under a phase-contrast microscope for the presence of intact intestinal crypts and lack of villi. The fractions with intact ileum crypts were pooled together and filtered through a 70 µm cell strainer (BD Biosciences #352350) into a 50 ml FBS-pre-coated tube to remove any debris. To remove single cell contamination from the heavier epithelial crypts, the crypts were pelleted by centrifuged at 80 g for 5 min at 4 °C. The supernatant was aspirated and crypt pellet was resuspended in 1–2 ml wash buffer, 100 µl was placed on a petri dish and the crypts were counted under a phase-contrast microscope.

**Ileum organoid culture**. To culture ileum organoids, a total of ~500 freshly isolated ileum crypts per well were mixed with 50 µl basement membrane matrix growth factor reduced Matrigel (BD Biosciences #356231) containing 1 µM Jagged-1 peptide (Notch Agonist) (AnaSpec #AS-61298) and plated in 24-well plates. The Matrigel was polymerized for 10 min at 37 °C incubator and 500 µl WENR growth media was added on top of the Matrigel. The ENR growth medium was prepared by mixing basal culture medium (BCM) [advanced DMEM/F-12 supplemented with 100 U/ml penicillin/100 µg/ml streptomycin, 50 µg/ml Gentamicin, 2 mM GlutaMAX, 10 mM HEPES (Gibco #15630-080), 1 mM N-acetyl-L-cysteine (Sigma-Aldrich #A9165), 1X B-27 Supplement (Gibco #17504-044), 1X N-2 Supplement (Gibco #17502-048), and 1% BSA (Gibco #15260-037)] with WNR conditioned medium in a 1:1 ratio, and 50 ng/ml murine Epidermal Growth Factor (mEGF) (Gibco #PMG8041). To maximize early growth of developing primary organoids from crypts, the WENR growth medium was supplemented with 10 µM Rho-associated protein kinase (ROCK) inhibitor, Y-27632 (Sigma-Aldrich #Y0503), 0.5 µM transforming growth factor (TGF)-β type I receptor inhibitor, and A-83-01 (R&D systems #2939) for the first 2 days of the culture. The culture medium was replaced every 2 days with ENR growth medium. For mRNA analysis

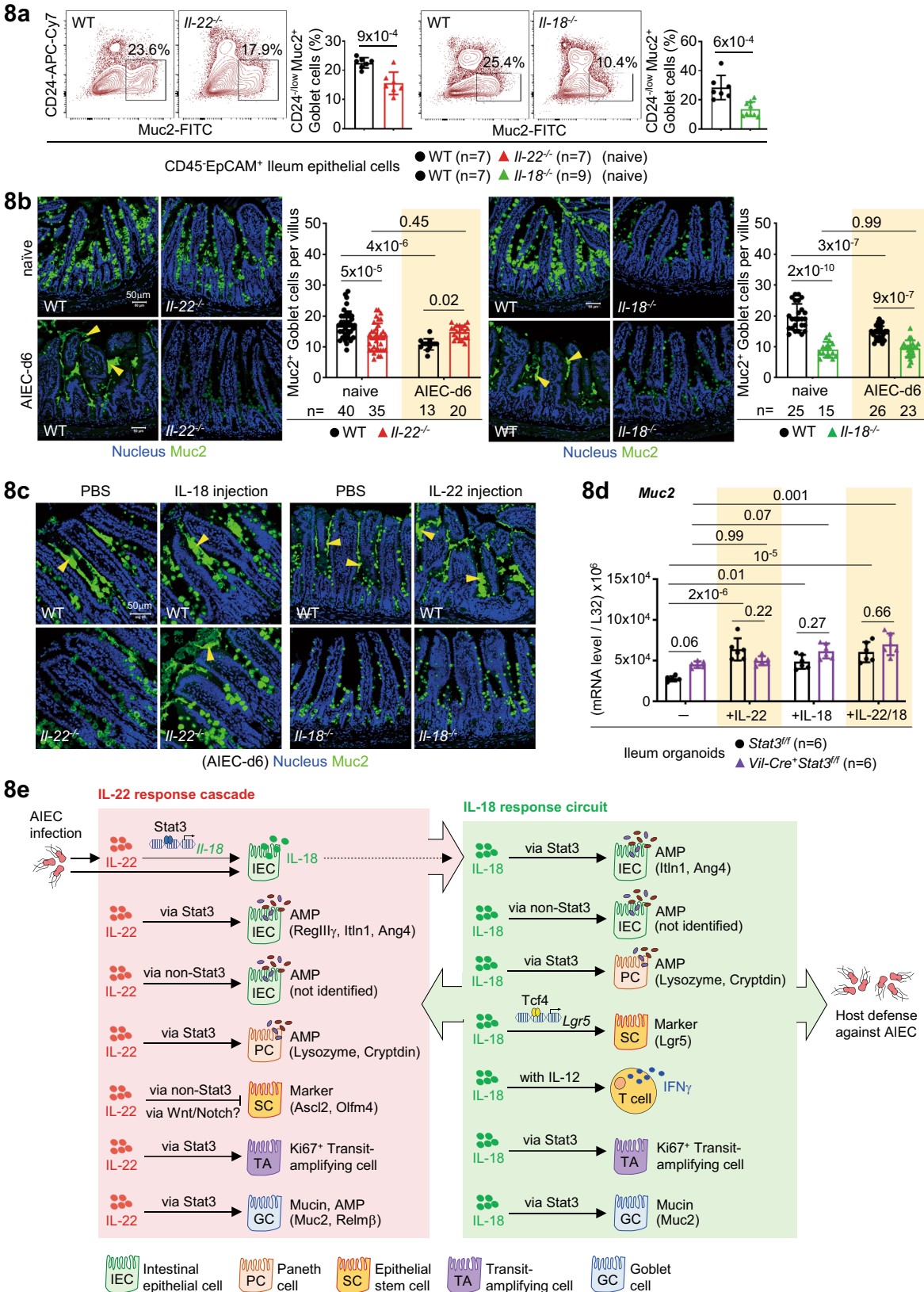

of stimulated organoids, some wells were treated with 50 ng/ml recombinant mouse IL-22 protein (eBioscience #14-8221-63), 100 ng/ml recombinant mouse IL-18 protein (R&D #9139-IL), or both, for 24 hr on day-5. For organoids morphology observation or protein analysis with FACS or immunofluorescence, the treatment wells were given recombinant IL-22 or IL-18 as above at the initial stage of organoid culture, and the medium was replaced every 2 days with fresh ENR growth medium contained fresh IL-22 or IL-18.

**Organoid observation and isolation from the Matrigel.** To observe the morphology of organoids, the 24-well plates containing organoids were observed by a phase-contrast microscope. Images were obtained with a Leica model DMi8 imaging system under 10x, 20x objectives with Leica Application Suite X software. The number of organoid buds was determined by counting new buds except for the original one. The size of the organoid was calculated from the longest length within the original bud. To remove the organoid from Matrigel, the growth medium was

**Fig. 8 IL-22 promotes mucin secretion via IL-18 during AIEC infection. a** Flow cytometry analysis of ileum epithelial cells, isolated from the indicated naïve mice, for Muc2+ Goblet cells. **b, c** Immunofluorescence analysis of ileum villi for Muc2+ Goblet cells in the indicated naïve or AIEC-infected mice at d6 (n = 3 in each group). Mucin in the lumen after infection is also detected and indicated by yellow arrowheads. Quantification of Muc2+ Goblet cells per villus is indicated. **d** Quantitative real-time PCR analysis of IL-22/IL-18-stimulated ileum organoids, derived from the indicated mice, for Goblet cell marker Muc2. **e** Graphic illustration of integrated and differential regulation of barrier function and epithelial subsets by IL-22 and IL-18 during AIEC host defence. IL-22 initiates an IL-18-dependent response circuit which boosts innate immunity and activates adaptive immunity. Each symbol in bar graphs represents an ileum sample (**a**) or organoid culture (**d**) derived from one mouse. Data shown are representative (**b**, **c**) or combined (**a**, **d**) results from two independent reproducible experiments. Statistical significance is indicated using unpaired two-tailed t test (**a**) or Two-way ANOVA with Tukey's multiple comparisons test (**b**, **d**). Data are presented as mean ± SD. Source data are provided as a Source Data file.

removed and Matrigel containing 6 days old organoids were washed with cold PBS twice. About 500 μl Cell Recovery Solution (CRS) (Corning #354253) was added to each well and the Matrigel was scrapped into an ice-cold 50 ml falcon tube. The well was additionally rinsed with CRS once and cell suspension was transferred to the same tube and shaken at 100 rpm on ice for 1 hr until the Matrigel was completely dissolved. After the organoids were released from the Matrigel, the tube was centrifuged at 300 g for 5 min at 4 °C and the supernatant was discarded. The organoid pellet was resuspended in 10 ml wash buffer and transferred to a 15 ml falcon tube and centrifuged at 20 g for 5 min at room temperature to remove any dead cells.

**Crypt killing assay.** Intestinal crypts were isolated as previously described[30]. Briefly, mice were euthanized, ileum or colon fragments were isolated, opened longitudinally, and immediately washed with ice-cold PBS. The fragments were immersed in PBS with 30 mM EDTA for 20 min on ice. After removal of EDTA buffer, the fragments were vigorously suspended by using a 3 ml dropper with cold PBS and briefly centrifuged at 200 g for 3 min. After discarding the supernatant, the resulting sediment, mostly containing the villi, was resuspended with PBS. After further vigorous resuspension, centrifugation, and filter through a 70 μm cell strainer (BD Bioscience), the supernatant was enriched for crypts. Isolated crypts were manually counted under a microscope. About 2000 crypts from each sample were in vitro stimulated with 100 ng/ml IL-22, 100 ng/ml IL-18, or both, to release anti-microbial peptides (AMP) at 37 °C for 30 min. AMP-containing supernatants were then incubated with 1000 CFU of live adherent-invasive E. coli (AIEC) for 90 min, and the AIEC titer after AMP-mediated killing were determined by CFU assay. The percentage of AIEC killing by crypts was normalized to those without cytokine stimulation.

**Chromatin immunoprecipitation (ChIP).** For ChIP assay in human HT-29 cells, $1.5 \times 10^7$ cells were stimulated with or without 50 ng/ml IL-22 for 15 min at 80% cell confluency. For ChIP assay in mouse CMT93 cells, $1.5 \times 10^7$ cells were stimulated with 100 ng/ml IL-18 and/or 5 μM MK-2206 (Akt inhibitor) for 24 h at 80% cell confluency. Cross-linking were performed with 1% formadehyde for 10 min at room temperature, followed by a quenching of formadehyde using 0.125 M glycine. Cells were washed with PBS 3 times. Nuclei were isolated in 400 μl of cell lysis buffer (10 mM Tris, pH 8.0, 10 mM NaCl, 0.2% NP-40) containing 1X phosphatase and protease inhibitor cocktail (MedChemExpress, HY-K0022 and HY-K0010) in the cold room (4 °C) with 30 min rotation. Nuclei were then centrifuged at 13,200 rpm in a refrigerated microcentrifuge and were lysed in 100 μl of nuclear lysis buffer (50 mM Tris, pH 8.1, 10 mM EDTA, 1% SDS, 1X phosphatase and protease inhibitor cocktail) in the cold room with 2 hr rotation. 1 ml of immunoprecipitation (IP) dilution buffer (20 mM Tris, pH 8.1, 2 mM EDTA, 150 mM NaCl, 1% Triton X-100, 0.01% SDS) was added and the chromatin was sheared with 14-16 pulse at 40% amplitude using an Ultrasonic Processor (110 V) with a 3 mm microtip probe (Misonix, XL-2020), with tubes immersed in cold water. Each pulse cycle is comprised of 5 sec sonication followed by 15 sec cool down. The chromatin was sheared to an average size of 200-1000 base pairs. Pellets were centrifuged at 13,200 rpm at 4 °C and the supernatant was mixed with 40 μl of packed protein A/G agarose beads (Santa Cruz Biotechology #sc-2003) for pre-clean process before immunoprecipitation. Supernatant was then split into three aliquots after centrifuged at 3,000 rpm and 700 μl of IP dilution buffer was added to each qliquot. 2.5 μg of antibody were added and the tubes were rotated overnight at 4 °C. 40 μl of packed protein A/G agarose beads were added and the samples were rotated for 2 hr at 4 °C. The beads were washed with IP wash I buffer (20 mM Tris, pH 8.1, 2 mM EDTA, 50 mM NaCl, 1% Triton X-100, 0.1% SDS) three times for 10 min each at 4 °C, followed by one wash with IP wash II buffer (10 mM Tris, pH 8.1, 1 mM EDTA, 0.25 M LiCl, 1% NP-40, 1% deoxycholic acid) for 10 min at 4 °C. Three washes with cold TE buffer (50 mM Tris, pH 8.0, 10 mM EDTA) were performed. After eluting twice using 200 μl of fresh elution buffer (100 mM sodium bicarbonate, 1% SDS) for 10 min at room temperature, 16 μl of 5 M NaCl were added and the cross-links were reversed at 65 °C overnight. The DNA was purified using PCR purification kit (Qiagen, #28106). 50 μl of each ChIP product was recovered and 1 μl was used for PCR. PCR was performed using following conditions: 95 °C for 30 sec, 55 °C for 30 sec, and 72 °C for 30 sec, for 35 cycles. Primer sequences are listed in the section of Supplementary Methods. After electrophoresis

analysis, the intensity of each PCR product band was quantified by ImageJ (1.53k) software. For ChIP assay in mouse ileum organoids, one 24-well plate of organoid culture was used. Organoids were isolated from the ileum and cultured (600 crypts for a well) for 5 days before stimulation with 50 ng/ml IL-22 or 100 ng/ml IL-18 for 24 h. Stimulated organoids were isolated from the the Matrigel and the ChIP assay was performed as described above. Transcription factor binding sites on the Il-18 or Lgr5 promoter (human and mouse) were predicted by database JASPAR. The PCR primers for ChIP assays were designed by Primer3web.

**Statistics.** All statistical analyses were performed using GraphPad Prism, v8.02 software. The results are expressed as the mean ± S.D. Statistical significance (p value) is indicated. Unpaired two-tailed t test, One-way ANOVA with Sidak's multiple comparisons test, or Two-way ANOVA with Tukey's multiple comparisons test is used for the analysis and indicated in each figure legend. The definition of symbols or sample numbers (n) is provided in each figure for statistics analysis.

**Reporting summary.** Further information on research design is available in the Nature Research Reporting Summary linked to this article.

## Data availability

The Source Data, containing all raw data presented in each figure of the main manuscript and supplementary information, as well as all uncropped Western blots and DNA gel blots, are provided with this paper. Transcription factor binding sites, illustrated in Supplementary Fig. 3a, 3c, and 8c, are predicted by the open-access database JASPAR (https://jaspar.genereg.net). The PCR primers used for all ChIP assays are designed by Primer3web (https://primer3.ut.ee/). All relevant primer sequences are listed in Supplementary information. Source data are provided with this paper.

## Code availability

Not applicable in this manuscript. Source data are provided with this paper.

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

## Acknowledgements

This work was supported by Career Development Award (#104-CDA-L07 to J.W.S.) from Academia Sinica, Taiwan, and by MOST grants (106-2320-B-001-009-MY3 and 106-2321-B-016-003 to J.W.S) from Ministry of Science and Technology (MOST), Taiwan. We thank the Pathology Core, Flow Cytometry Core, Confocal Microscopy Core, and DNA Sequencing Core facilities at the Institute of Biomedical Sciences (IBMS) of Academia Sinica, for their technical assistance. We thank the National RNAi Core Facility in Academia Sinica for providing shRNA reagents and related services. We thank Dr. Brian K. Coombes, Department of Biochemistry and Biomedical Sciences, McMaster University, Canada, for kindly providing the AIEC isolate NRG857c, Dr. Richard S. Blumberg, Department of Medicine, Brigham and Women's Hospital, Harvard Medical School, for kindly providing *Defa6-Cre* mice, and Dr. Wenjun Ouyang, Department of Immunology, Genentech, for kindly providing *Il-22⁻/⁻* mice.

## Author contributions

H.Y.C and H.H.L. designed and performed experiments, and contributed to manuscript writing and revision. J.N.S., Y.W.C., and N.S.S. helped perform experiments and data collection. Y.T.W. provided animal breeding and husbandry assistance. J.W.S. designed experiments and wrote the manuscript.

## Competing interests

The authors declare no competing interests.
