## [Peer Review File · Nature Communications]

IL-22 initiates an IL-18-dependent epithelial response circuit to enforce intestinal host defenceREVIEWER COMMENTS

Reviewer #1 (Remarks to the Author):

In this study Chiang et al. aimed to analyze the relation of IL-22 and IL-18 during host defense. To this end, they used an in vivo infection model with adherent-invasive E. coli (AIEC) as well as IL-22KO and IL-18KO mice. Furthermore, they used organoids to analyze the involved signaling pathways.

The authors show that IL-22 and IL-18 are induced in the intestine upon infection of mice with AIEC. IL-22KO and IL-18KO mice were more susceptible to infection and this was associated with reduced crypt cell proliferation and production of antimicrobial peptides. IL-18 levels and CD8+ IFN γ + T cells were reduced in IL-22 KO AIEC infected mice compared to WT mice. Furthermore, IFN γ KO mice were more susceptible to AIEC infection compared to WT mice. Thus, the authors claim a direct link of IL-22 - IL-18 - IFN γ and disease outcome in their model. Using ileum organoids and recombinant IL-22 and IL-18 they found that IL-22 and IL-18 are functionally linked and coordinated to upregulate anti-microbial immunity of Paneth cells. Finally, they found that IL-22 induces IL-18 via Stat3. Of note, IL-18 induced LGR5+ stem cells in a stat3 independent but TCF-4 dependent manner.

Overall the experiments appear to be well performed, and this area is of potential interest. However, the novelty of this study is partially compromised as it was shown before that IL-22 induces IL-18, which is protective in bacterial and viral infection (e.g. Munoz et al. Immunity 2015; Zhang et al. Science Immunol 2020). This study would still represent a significant extension of this work, but some of their conclusions as outlined below are not sufficiently supported by their data.

Major points:

Figure 3: The authors claim that "Collectively, we reveal an IL-22-initiated IL-18 response cascade, where IL-22 transcriptionally activates epithelial IL-18 by promoting phospho-Stat3 binding to the IL-18 promoter, leading to subsequent IL-18-mediated IFN induction preferentially in CD8+ T cells which are critical for AIEC clearance." However, direct in vivo evidence supporting this conclusion is missing. This is critical since IL-22KO mice show a higher disease compared to WT mice, which may bias the results. Thus, the authors would need to use IL-22 KO x IL-18 KO mice to show that the effects of IL-22 are indeed IL-18 dependent. Alternatively, the authors could overexpress IL-18 in IL-22KO.

Figure 4: The authors conclude that: "...it is IL-22-initiated IL-18 signaling that is required for mounting T-cell IFN response for AIEC host defense." However, also here direct in vivo evidence supporting this finding is missing. The results are correlative in nature and may be biased due to differences in disease severity. The authors should provide direct evidence linking IL-22 -> IL-18 -> IFN γ producing CD8 T cells and disease outcome.

General points:

The authors should indicate, if they used littermate controls for all their experiments. This is key since several of the used mouse strains are known to have an altered microbiota.

The authors used mostly an unpaired two tailed t-test. However, this is not appropriate in case of multiple comparison. In this case an ANOVA test and a post hoc test would be more appropriate.

Minor points:

Figure 2 and Figure 4: The author should show IL-22 (Figure 2) and IL-18 protein (Figure 4) in addition to RNA expression.

Reviewer #2 (Remarks to the Author):

This study Chiang HY et al focuses on understanding the role of IL-22-dependent and independent IL-18-mediated epithelial and stem cell host defense in the intestine under homeostatic and after pathogen (AIEC) infection. The authors explore the potential differential and synergistic role of IL-22 and IL-18 in regulating stem cell and Paneth cell function. The authors demonstrate that AIEC infection triggers stem and Paneth cell-dependent regenerative and antimicrobial program, respectively. They further show that epithelial regenerative and Paneth cell antimicrobial functions are dependent on IL-22 driven IL-18 responses. The studies were elegantly done with high quality data. However, some of the IL-22 related observations (IL-18 and Paneth cell AMPs expression) and IL-12+ IL-18-mediated IFN γ responses (in immune cells) are well-characterized in prior studies. Further validation of these work by authors is appreciated but novelty is impacted. There are several novel observations: 1) Paneth cell are a direct target of IL-22, which is consistent with a very recent literature (PMID33060802), 2) IL-18-mediated regulation of Paneth cells number, budding and antimicrobial functions, 3) Additive effects of IL-18 and IL-22 to enhance Paneth cells antimicrobial functions in vitro and in vivo, 4) Differential role (STAT3 dependent and independent) of IL-22 and IL-18 in regulating Lgr5+ ISC stemness, and 5) IL-18 promote TCF4 binding to the Lgr5 promoter. Below are my suggestions to help strengthen the manuscript.

Major comments

- 1) The PI showed that both IL-22 and IL-18 induced Paneth cell antimicrobial peptide expression equally. It would be interesting to know if IL-22-mediated Paneth cell antimicrobial and organoid growth responses are dependent on epithelial IL-18. Would neutralizing IL-18 affects IL-22-mediated Paneth cell responses?
- 2) In organoid experiments, the PI used Vil-cre;Stat3fl/fl mice in which STAT3 signaling was compromised in entire epithelium and ISCs. They further showed that IL-22 and IL-18 had differential impacts on Lgr5+ ISC. Why not use Defa6-cre;Stat3fl/fl mice as well for experiments related to Paneth cell antimicrobial activity and organoid growth/budding? The PI has access to these mice.
- 3) It would be useful to show IL-22Ra1 and IL-18R expression on Paneth cells. It is possible that IL-22 and IL-18 mediated effects were restricted to a different Paneth cell subsets.
- 4) The PI showed that Lgr5+ ISC and Paneth cell number and functions were compromised in naive and AIEC infected Il22^{-/-} and Il18^{-/-} mice. There may be additional targets or mechanisms independent of ISC and Paneth cells. Direct relevance of Lgr5+ ISC or Paneth cells in the AIEC model is not established in current study. It would strengthen the manuscript if the role of Defa6-cre;Stat3fl/fl mice were investigated in AIEC infection.
- 5) Organoid growth and budding data in response to IL-22 and IL-18 stimulation (24 hours) is interesting. However, the cytokines dose (50ng/ml and 100ng/ml) seems very high and the treatment regime is limited to 24 hours given prior literature on the effect of continuous stimulation of IL-22 on organoid growth. The manuscript would benefit from additional description of the choice treatment time (24 hours) and dose. Could the author show that a lower dose of IL-18 has the same effect on organoid growth/budding?
- 6) The authors should provide survival and weight curves as well as a histological score of the mice (Il22^{-/-}, Il18^{-/-} and Vil-cre;IL-22Ra1fl/fl) infected with AIEC.
- 7) Since IL-18 regulates Lyz1 and Cryptdin expression at transcripts level, it would be useful to confirm the loss of Paneth cells in AIEC infected Il18^{-/-}-mice by other methods (UEA1, alcian blue or phloxine-tatrazine).
- 8) The expression level of Dll1, Dll4, Atoh1 and Hes1 needs to be shown in Il18^{-/-} mice at steady state and after infection.

Minor

- The major source of IL-22 is ILC3 in the intestine. This needs to be included in the introduction (second paragraph).
- Muc2 antibody source is missing
- What are the mechanisms of how IL-18R signaling promotes TCF4 nuclear translocation? If the author has any speculation, please discuss.
- It would be useful to readers if the author includes discussion of their Paneth cell related findings with PMID30364840, PMID33060802, reference 22 and 24.
- It is better to move figure 3C-D to supplemental.
- It appears that IL-22-mediated STAT3-dependent organoid growth is independent of Claudin and Lgr5+ ISC. How does IL-22 promote organoid growth? If authors have any speculations, please discuss.

Reviewer #3 (Remarks to the Author):

This is a potentially interesting study describing cooperativity between IL22 and IL18 during E coli infection. However, it is difficult to draw strong conclusions from the data presented. It is not convincing that the antimicrobial molecules attributed to Paneth cells are actually coming from Paneth cells. It is also unclear if changes in Paneth cell and Goblet cell frequencies are due to impacts on those cells or to impacts on their precursors, including ISCs. Please find below additional specific comments.

Major comments:

- Goblet cells can be a source of Reg3y, Ang4, and others. To confirm a direct effect of IL-18 and IL-22 on Paneth cells, it would be more convincing to treat sorted paneth cells with IL-22 or possibly to gavage Defa6-il22r1 f/f mice with AIEC and check for changes in antimicrobial activity.

- Authors suggest that IL-22 induces the upregulation of RegIIIy, Relmb, Itln1, and Ang4, all with anti-gram-negative. They also state that IL-18 can also induce Itln1 and Ang4. Because IL-18 is also induced by IL-22, they cannot exclude that Itln1 and Ang4 are not directly upregulated by IL-22 but indirectly through IL-18. To test this hypothesis they need to treat IL-18 KO organoids with IL-22 and check for gene changes. In the alternative, the fact that IL-18 can only induce some antimicrobial suggests that IL-22 and IL-18 may be targeting distinct cells types.

- Fig2 H 2I. Data does not exclude the possibility that IL-22/STAT3 AXIS is acting on precursor secretory cells promoting differentiation rather than act on fully differentiated paneth cells. Is the reduction in lysozyme positive cells the results of increased apoptosis in adult paneth cells or reduced differentiation from precursors? Apoptotic markers, for example tunnel assay, together with lysozyme stain is required.

Minor

- Fig 1F. The authors show a western blot with two bands but only one changes. Have the authors confirmed which band is LGR5?

- S1B why are crypts from WT mice so small and fragmented compared to Defa6-Cre+-Rosa mice?

- Fig 2A: increase in mRNA for IL-22 is not always correlated with increased production and release of IL-22. At least ELISA should be done to test the amount of protein.

- Fig 2B: staining of Ki67 excludes the nuclear region and is mainly in the cytoplasm, but normally Ki67 is a nuclear marker. Cells in the crypts regions have less Ki67 but the total number appears to be unchanged. Quantification of the average number of Ki67+ cells in each crypt should be performed.

- Fig 2E and 2F: How do the authors explain the discrepancy between the in vitro and vivo findings where they observed the opposite results in absence of stat3? In homeostasis after depletion of stat3, Itln1 and Ang4 go up in organoids but go down (Itln1) or unchanged (Ang4) in ileum crypts. In organoids, the depletion of Stat3 induces no changes in Relmb, but crypts show a dramatic reduction in Relmb after depletion.

- Fig 4G -4F: FACS data and immunofluorescence suggest that depletion of IL-18 alone is sufficient for changes in paneth cell numbers. Can the loss of paneth cells be rescued in IL-18 -/- mice by injecting IL-22 after AIEC gavage?

- Evaluation for evidence of direct phosphorylation of STAT3 in Paneth cells by IL-22 and IL-18 is necessary. In addition, it would be useful to evaluate the total amount of p-stat3 in goblet cells after treatment with IL-22 and IL-18.

This is the rebuttal letter for the manuscript (NCOMMS-20-42875), "**IL-22 initiates an IL-18 epithelial response circuit for host defense boost**". Our point-to-point response (shown in black color) to each concern raised by reviewers (shown in green color) is listed below.

Reviewer-1: (Remarks to the Author)

In this study Chiang et al. aimed to analyze the relation of IL-22 and IL-18 during host defense. To this end, they used an in vivo infection model with adherent-invasive E. coli (AIEC) as well as IL-22KO and IL-18KO mice. Furthermore, they used organoids to analyze the involved signaling pathways. The authors show that IL-22 and IL-18 are induced in the intestine upon infection of mice with AIEC. IL-22KO and IL-18KO mice were more susceptible to infection and this was associated with reduced crypt cell proliferation and production of anti-microbial peptides. IL-18 levels and CD8+ IFN γ + T cells were reduced in IL-22 KO AIEC infected mice compared to WT mice. Furthermore, IFN γ KO mice were more susceptible to AIEC infection compared to WT mice. Thus, the authors claim a direct link of IL-22 - IL-18 - IFN γ and disease outcome in their model. Using ileum organoids and recombinant IL-22 and IL-18 they found that IL-22 and IL-18 are functionally linked and coordinated to upregulate anti-microbial immunity of Paneth cells. Finally, they found that IL-22 induces IL-18 via Stat3. Of note, IL-18 induced LGR5+ stem cells in a stat3 independent but TCF-4 dependent manner. Overall the experiments appear to be well performed, and this area is of potential interest. However, the novelty of this study is partially compromised as it was shown before that IL-22 induces IL-18, which is protective in bacterial and viral infection (e.g. Munoz et al. *Immunity* 2015; Zhang et al. *Science Immunol* 2020). This study would still represent a significant extension of this work, but some of their conclusions as outlined below are not sufficiently supported by their data.

Our response:

We thank the reviewer's positive comments on our study. In addition to those novel observations mentioned above by this reviewer, we now provide more novel results in the revised manuscript and hopefully the improvement will show higher impacts and distinguish our studies from prior IL-22/IL-18-related publications.

Our novel and updated findings include ...

- (1) Genetic evidence to show Paneth cells (PC) (revised **Fig.1i**) and Stat3 in PC (revised **Fig.2i, 4j-m**) are crucial for AIEC host defense which is highly relevant to Crohn's disease.
- (2) Genetic evidence to show Stat3 in Paneth cells is the direct target of IL-22 and IL-18, by IL-22/IL-18-stimulated Defa6-Cre⁺Stat3^{F/F} organoids (revised **Fig.2f, s1f, 4c, s4d**). IL-22-Stat3 and IL-18-Stat3 signaling in PC is crucial for the production of PC-specific anti-microbial products (revised **Fig.s1g, 4b**).
- (3) IL-18 failed to rescue the susceptibility (by CFU assay) of Defa6-Cre⁺Stat3^{F/F} mice to AIEC infection (revised **Fig.4j**), indicative of a key role of IL-18-Stat3 signaling in Paneth cells during host defense.
- (4) IL-22 or IL-18 induces Stat3 phosphorylation in Lysozyme⁺ Paneth cells and Muc2⁺ Goblet cells in vivo (revised **Fig.4d, s4e**).
- (5) Three sets of rescue experiments provide in vivo evidence to support the IL-22-IL-18-IFN γ axis and IL-22-IL-18-Paneth cell axis are crucial for AIEC host defense (revised **Fig.4j-m, 5a-f**).
- (6) An unique and novel role for IL-18 to promote organoid budding (revised **Fig.6e-f**) and Lgr5⁺ stem cell growth (revised **Fig.7e,7g**).
- (7) IL-18, but not IL-22, specifically induces an Akt-TCF4-Lgr5 pathway to promote stem cells by inducing TCF4 binding to the P1 region within mouse Lgr5 promoter (revised **Fig.7f-h**).
- (8) Both IL-22 and IL-18 promotes luminal mucin secretion during AIEC infection (revised **Fig.8d**). IL-18 injection restores mucin secretion in IL-22^{-/-} mice while IL-22 fails to do so in IL-18^{-/-} mice (revised **Fig.8e**).

We have also cited and discussed six very related prior publications that might impact our study. In brief, while it was shown that IL-22 induces IL-18, which is protective in bacterial and viral infection (**Munoz et al. *Immunity* 2015; Zhang et al. *Science Immunol* 2020**), a role for IL-22 or IL-18 was not explored in Paneth cells in the context of bacterial and viral infection. We identified and provide new evidence to show IL-22 and IL-18 are a bona fide regulator of stem cells and Paneth cells in vitro and in vivo during AIEC infection. Similarly, while a

role for IL-22 in promoting Lysozyme⁺ Paneth cells was shown and discussed in **Zwarycz et al. Cell Mol Gastroenterol Hepatol 7:1, 2019**; and **Zha et al. Cell Mol Gastroenterol Hepatol 7:255, 2019**, both papers did not have data about IL-18 in stem cells and Paneth cells. Furthermore, while **Lindemans et al. Nature 528:560, 2015** showed that IL-22 does not enhance Paneth cell frequency or STAT3 phosphorylation in vitro (Fig.s7), they did not have data about IL-18 in stem cells and Paneth cells but we now provide new evidence that IL-22 and IL-18 could induce Stat3 phosphorylation in Lysozyme⁺ Paneth cells and Muc2⁺ Goblet cells in vivo. While **Gaudino et al. Mucosal Immunol 14:389, 2020** showed IL-22 receptor signaling in Paneth cells is critical for their maturation and anti-Salmonella immunity, they did not have data about a role for IL-18 in Paneth cells or stem cells. Therefore, our comprehensive study on the role for IL-22 and IL-18 in stem cells, TA cells, Paneth cells, and Goblet cells in the context of AIEC infection provides more novelty and represents a significant extension to those prior publications.

Major points:

Figure 3: The authors claim that "Collectively, we reveal an IL-22-initiated IL-18 response cascade, where IL-22 transcriptionally activates epithelial IL-18 by promoting phospho-Stat3 binding to the IL-18 promoter, leading to subsequent IL-18-mediated IFN induction preferentially in CD8⁺ T cells which are critical for AIEC clearance." However, direct in vivo evidence supporting this conclusion is missing. This is critical since IL-22 KO mice show a higher disease compared to WT mice, which may bias the results. Thus, the authors would need to use IL-22 KO x IL-18 KO mice to show that the effects of IL-22 are indeed IL-18 dependent. Alternatively, the authors could overexpress IL-18 in IL-22KO.

Our response:

We thank the reviewer to point out this, which we also consider as an important mechanistic data that is missing in our previous manuscript. To address this, we have performed three sets of rescue experiments (total six independent exp) to dissect whether IL-22 regulates IFN γ via IL-18 (or a direct IL-22-IL-18-IFN γ axis).

Rescue Exp-1 (revised **Fig.5a-c**, data shown are combined results from two independent exp):

As noted in revised **Fig.5a**, IL-18 injection into IL-22 KO mice indeed rescue the susceptibility of IL-22 KO to AIEC, by reducing CFU in most intestinal samples to those levels of WT mice and by increasing IFN γ ⁺ CD4⁺ T (%+MFI) and CD8⁺ T (%) cells in the lamina propria (revised **Fig.5b**). Furthermore, in revised **Fig.5c**, while IL-18 injection into WT mice did not further upregulate lysozyme during AIEC infection, injection of IL-18 into IL-22 KO mice significantly promotes lysozyme levels (or Lysozyme⁺ Paneth cells), which is a very specific marker of Paneth cells (revised **Fig.s1c**). Upregulation of IFN γ ⁺ T cells and Lysozyme⁺ Paneth cells in IL-18-injected IL-22 KO mice (as suggested by the reviewer: to overexpress IL-18 in IL-22 KO) provides in vivo evidence that IL-22 indeed regulate IFN γ via IL-18, or support a direct link of IL-22-IL-18-IFN γ axis for AIEC infection.

Rescue Exp-2 (revised **Fig.5d-f**, data shown are representative results from two independent exp):

To further support the above conclusion, we performed a reciprocal rescue exp to test whether IL-22 injection into IL-18 KO mice could rescue the susceptibility of IL-18 KO to AIEC. As noted in revised **Fig.5d**, IL-22 injection into IL-18 KO mice clearly failed to reduce CFU in most intestinal samples to those levels of WT mice. Correlated to this, IL-22 injection also failed to upregulate IFN γ ⁺ CD4⁺ T and CD8⁺ T cells in the lamina propria (revised **Fig.5e**). Furthermore, in revised **Fig.5f**, injection of IL-22 into IL-18 KO mice can not upregulate lysozyme levels in IL-18 KO. Together, these results further support a direct IL-22-IL-18-IFN γ axis, where IL-22 is a bona fide upstream regulator of IL-18 leading to IFN γ production in T cells and Paneth cell upregulation. These results also indicate a dominant requirement for IL-18 over IL-22 in promoting Paneth cell numbers and function during AIEC infection.

Rescue Exp-3 (revised **Fig.4j~4m, s4l**, data shown are combined results from two independent exp):

To test the key role of IL-18 in Lysozyme⁺ Paneth cells in vivo and to further support the above two rescue exp (revised **Fig.5c** and **5f**), we performed the third rescue exp to test whether IL-18 injection into Defa6-Cre⁺Stat3^{F/F} mice could rescue the susceptibility of Paneth cell-specific Stat3 conditional KO to AIEC. As noted in revised

Fig.4j, IL-18 injection into Defa6-Cre⁺Stat3^{F/F} mice failed to reduce CFU in all intestinal samples to those levels of WT mice. Injection of IL-18 into infected Defa6-Cre⁺Stat3^{F/F} mice also failed to fully upregulate lysozyme⁺ Paneth cells (revised **Fig.4k-l**) and mRNA levels of Paneth cell-related products (Lysozyme/Cryptdin in revised **Fig.4m** and Itln1/Ang4 in revised **Fig.s4l**).

Figure 4: The authors conclude that: '...it is IL-22-initiated IL-18 signaling that is required for mounting T-cell IFN response for AIEC host defense.' However, also here direct in vivo evidence supporting this finding is missing. The results are correlative in nature and may be biased due to differences in disease severity. The authors should to provide direct evidence linking IL-22 -> IL-18 -> IFN γ producing CD8 T cells and disease outcome.

Our response:

We thank the reviewer to point out this and we have performed three sets of rescue exp (total six independent exp) to fully address this question. We also want to point out that as the protocol for these rescue exp is short (outlined in revised **Fig.s5a** and the analysis was performed **Fig.s4h,s8b**), there were basically no detectable macroscopic pathology (revised **Fig.s4h,s8b**), or altered survival rate, during the six-day course of AIEC infection. The only disease outcome that can be convincingly quantified in this acute infection model is CFU measurement of feces and intestinal tissues, as shown in revised **Fig.4j, 5a, and 5d**.

General points:

The authors should indicate, if they used littermate controls for all their experiments. This is key since several of the used mouse strains are known to have an altered microbiota.

Our response:

We thank the reviewer to point out this fundamental question. Yes, littermate mice (littermate WT vs IL-22^{-/-} or IL-18^{-/-}, littermate Stat3^{F/F} vs Vil-Cre⁺Stat3^{F/F} or Defa6-Cre⁺Stat3^{F/F} mice) were used in all experiments. Wild-type controls were generated by intercrossing of heterozygous KO mice. We have updated the information in the section of "Materials and Methods".

The authors used mostly an unpaired two tailed t-test. However, this is not appropriate in case of multiple comparison. In this case an ANOVA test and a post hoc test would be more appropriate.

Our response:

We thank the reviewer to point out this concern, as the use of inappropriate statistical methods could lead to different conclusions. Because of this, we have carefully re-done all data analyses where we specifically indicate "Unpaired two-tail t-test", 'One-way ANOVA with Dunnett's multiple comparisons test', or "Two-way ANOVA with Tukey's multiple comparison test". We have updated the information in the section of "Statistics".

Minor points:

Figure 2 and Figure 4: The author should show IL-22 (Figure 2) and IL-18 protein (Figure 4) in addition to RNA expression.

Our response:

We now provide both protein and mRNA measurements of IL-22 (revised **Fig.2a**) and IL-18 (revised **Fig.4a**).

Reviewer-2: (Remarks to the Author)

This study Chiang HY et al focuses on understanding the role of IL-22-dependent and independent IL-18-mediated epithelial and stem cell host defense in the intestine under homeostatic and after pathogen (AIEC) infection. The authors explore the potential differential and synergistic role of IL-22 and IL-18 in regulating stem cell and Paneth cell function. The authors demonstrate that AIEC infection triggers stem and Paneth cell-dependent regenerative and antimicrobial program, respectively. They further show that epithelial regenerative and Paneth cell antimicrobial functions are dependent on IL-22 driven IL-18 responses. The studies were elegantly done with high quality data. However, some of the IL-22 related observations (IL-18 and Paneth cell AMPs expression) and IL-12+ IL-18-mediated IFN γ responses (in immune cells) are well-characterized in prior studies. Further validation of these work by authors is appreciated but novelty is impacted. There are several novel observations: 1) Paneth cell are a direct target of IL-22, which is consistent with a very recent literature (PMID33060802), 2) IL-18-mediated regulation of Paneth cells number, budding and antimicrobial functions, 3) Additive effects of IL-18 and IL-22 to enhance Paneth cells antimicrobial functions in vitro and in vivo, 4) Differential role (STAT3 dependent and independent) of IL-22 and IL-18 in regulating Lgr5⁺ ISC stemness, and 5) IL-18 promote TCF4 binding to the Lgr5 promoter. Below are my suggestions to help strengthen the manuscript.

Our response:

We thank the reviewer's positive comments on our study. In addition to those novel observations mentioned above by this reviewer, we now provide more novel results in the revised manuscript and hopefully the improvement will show higher impacts and distinguish our studies from prior IL-22/IL-18-related publications.

Our novel and updated findings include ...

- (1) Genetic evidence to show Paneth cells (PC) (revised **Fig.1i**) and Stat3 in PC (revised **Fig.2i, 4j-m**) are crucial for AIEC host defense which is highly relevant to Crohn's disease.
- (2) Genetic evidence to show Stat3 in Paneth cells is the direct target of IL-22 and IL-18, by IL-22/IL-18-stimulated Defa6-Cre⁺Stat3^{F/F} organoids (revised **Fig.2f, s1f, 4c, s4d**). IL-22-Stat3 and IL-18-Stat3 signaling in PC is crucial for the production of PC-specific anti-microbial products (revised **Fig.s1g, 4b**).
- (3) IL-18 failed to rescue the susceptibility (by CFU assay) of Defa6-Cre⁺Stat3^{F/F} mice to AIEC infection (revised **Fig.4j**), indicative of a key role of IL-18-Stat3 signaling in Paneth cells during host defense.
- (4) IL-22 or IL-18 induces Stat3 phosphorylation in Lysozyme⁺ Paneth cells and Muc2⁺ Goblet cells in vivo (revised **Fig.4d, s4e**).
- (5) Three sets of rescue experiments provide in vivo evidence to support the IL-22-IL-18-IFN γ axis and IL-22-IL-18-Paneth cell axis are crucial for AIEC host defense (revised **Fig.4j-m, 5a-f**).
- (6) An unique and novel role for IL-18 to promote organoid budding (revised **Fig.6e-f**) and Lgr5⁺ stem cell growth (revised **Fig.7e,7g**).
- (7) IL-18, but not IL-22, specifically induces an Akt-TCF4-Lgr5 pathway to promote stem cells by inducing TCF4 binding to the P1 region within mouse Lgr5 promoter (revised **Fig.7f-h**).
- (8) Both IL-22 and IL-18 promotes luminal mucin secretion during AIEC infection (revised **Fig.8d**). IL-18 injection restores mucin secretion in IL-22^{-/-} mice while IL-22 fails to do so in IL-18^{-/-} mice (revised **Fig.8e**).

We have also cited and discussed six very related prior publications that might impact our study. In brief, while it was shown that IL-22 induces IL-18, which is protective in bacterial and viral infection (**Munoz et al. Immunity 2015; Zhang et al. Science Immunol 2020**), a role for IL-22 or IL-18 was not explored in Paneth cells in the context of bacterial and viral infection. We identified and provide new evidence to show IL-22 and IL-18 are a bona fide regulator of stem cells and Paneth cells in vitro and in vivo during AIEC infection. Similarly, while a role for IL-22 in promoting Lysozyme⁺ Paneth cells was shown and discussed in **Zwarycz et al. Cell Mol Gastroenterol Hepatol 7:1, 2019**; and **Zha et al. Cell Mol Gastroenterol Hepatol 7:255, 2019**, both papers did not have data about IL-18 in stem cells and Paneth cells. Furthermore, while **Lindemans et al. Nature 528:560, 2015** showed that IL-22 does not enhance Paneth cell frequency or STAT3 phosphorylation in vitro (Fig.s7), they did not have data about IL-18 in Paneth cells and we now provide new

evidence that IL-22 and IL-18 could induce Stat3 phosphorylation in Lysozyme⁺ Paneth cells and Muc2⁺ Goblet cells in vivo. While **Gaudino et al. Mucosal Immunol 14:389, 2020** showed IL-22 receptor signaling in Paneth cells is critical for their maturation and anti-Salmonella immunity, they did not have data about a role for IL-18 in Paneth cells. Therefore, our comprehensive study on the role for IL-22 and IL-18 in stem cells, TA cells, Paneth cells, and Goblet cells in the context of AIEC infection provides more novelty and represents a significant extension to those prior publications.

Major comments

1) The PI showed that both IL-22 and IL-18 induced Paneth cell antimicrobial peptide expression equally. It would be interesting to know if IL-22-mediated Paneth cell antimicrobial and organoid growth responses are dependent on epithelial IL-18. Would neutralizing IL-18 affect IL-22-mediated Paneth cell responses?

Our response:

This is a very good question to test. To address this, we analyzed organoid growth in IL-22-stimulated IL-18^{-/-} organoids and found that both budding and size of organoids were not affected by epithelial IL-18 deficiency (revised **Fig.6h**). However, certain IL-22-induced anti-microbial products (RegIII γ / Relm β / Muc2, which are non-Paneth cell-specific, and Lysozyme / Cryptdin / Ang4, which are Paneth cell-specific or -related) were partially compromised by epithelial IL-18 deficiency (revised **Fig.s5b, 8g**). For in vivo rescue exp, injection of IL-22 into IL-18 KO mice failed to upregulate Lysozyme⁺ Paneth cells and rescue CFU or IFN γ during AIEC infection (revised **Fig.5d-f**). Injection of IL-22 into IL-18 KO mice failed to promote mucin secretion from Goblet cells during AIEC infection (revised **Fig.8e**). Of note, IL-22/IL-18 show additive effect on bacteria-killing by fresh crypts (revised **Fig.4e**), indicating that they have some non-redundant role in crypt immunity. Taken together, these results indicate that IL-22 and IL-18 show redundant and non-redundant effects, depending on the contexts which are being tested.

2) In organoid experiments, the PI used Vil-cre;Stat3^{fl/fl} mice in which STAT3 signaling was compromised in entire epithelium and ISCs. They further showed that IL-22 and IL-18 had differential impacts on Lgr5⁺ ISC. Why not use Defa6-cre;Stat3^{fl/fl} mice as well for experiments related to Paneth cell antimicrobial activity and organoid growth/budding? The PI has access to these mice.

Our response:

We very much agree with the reviewer's comments here and have therefore performed many in vitro and in vivo experiments to fully address this.

- (1) Paneth cell (PC)-deficient mice (with Defa6-Cre⁺Rosa-LSL-DTA mice) were more susceptible to AIEC infection (revised **Fig.1i, s1c**).
- (2) Induction of PC-specific anti-microbial products by IL-22 (revised **Fig.2f, s1f-g**), or by IL-18 (revised **Fig.4b-c, s4d**), was compromised in Defa6-Cre⁺Stat3^{F/F} organoids.
- (3) Defa6-Cre⁺Stat3^{F/F} mice were more susceptible to AIEC infection by CFU assay (revised **Fig.2i**).
- (4) Induction of epithelial IL-18 by IL-22 was reduced in Defa6-Cre⁺Stat3^{F/F} organoids (revised **Fig.3f**).
- (5) Injection of IL-18 failed to rescue the susceptibility of Defa6-Cre⁺Stat3^{F/F} mice to AIEC infection by CFU assay (revised **Fig.4j**).
- (6) Injection of IL-18 into infected Defa6-Cre⁺Stat3^{F/F} mice failed to upregulate Lysozyme⁺ Paneth cells (revised **Fig.4k-l**) and mRNA levels of Paneth cell-related anti-microbial products (Lysozyme/Cryptdin in revised **Fig.4m** and Itln1/Ang4 in revised **Fig.s4l**).
- (7) IL-22 or IL-18-induced organoid size was not affected in Defa6-Cre⁺Stat3^{F/F} organoids. However, IL-18 (but not IL-22)-induced organoid budding was compromised in Defa6-Cre⁺Stat3^{F/F} organoids (revised **Fig.s6b**). This is likely due to a negative feedback effect of niche (growth) factors from PC to support nearby stem cells. That is, functionality-compromised IL-18-stimulated Defa6-Cre⁺Stat3^{F/F} Paneth cells could not provide sufficient niche factors to fully support nearby stem cells for budding during culture.

3) It would be useful to show IL-22Ra1 and IL-18R expression on Paneth cells. It is possible that IL-22 and IL-18 mediated effects were restricted to a different Paneth cell subsets.

Our response:

This is a very good question to test. To address this, we first analyzed IL-22R (with an IL-22Ra1-specific antibody) and IL-18R (with an IL-18R1-specific antibody) expression in WT mice in the gated Paneth cells (CD24⁺ Lysozyme⁺), compared to CD24⁻Lysozyme⁻ and CD24^{low} Lysozyme⁻ subsets (revised **Fig.3g**). We also analyzed IL-22R and IL-18R expression in WT mice in the gated Goblet cells (CD24^{-/low} Muc2⁺) (revised **Fig.s8a**). As noted, we also compared IL-22R and IL-18R expression in the gated Paneth cells (CD24^{high} Lysozyme⁺) in WT vs IL-22/IL-18 KO mice and found that the expression levels of IL-22R or IL-18R in Paneth cells were not altered in both KO mice (revised **Fig.s2f**).

4) The PI showed that Lgr5⁺ ISC and Paneth cell number and functions were compromised in naïve and AIEC infected IL22^{-/-} and IL18^{-/-} mice. There may be additional targets or mechanisms independent of ISC and Paneth cells. Direct relevance of Lgr5⁺ ISC or Paneth cells in the AIEC model is not established in current study. It would strengthen the manuscript if the role of Defa6-cre;Stat3^{fl/fl} mice were investigated in AIEC infection.

Our response:

We very much agree with the reviewer's comments. To address this, we performed the following experiments.

(1) Paneth cell-deficient mice (with Defa6-Cre⁺Rosa-LSL-DTA mice) were more susceptible to AIEC infection by CFU assay (revised **Fig.1i, s1c**).

(2) Defa6-Cre⁺Stat3^{FF/FF} mice were more susceptible to AIEC infection by CFU assay (revised **Fig.2i**).

(3) Injection of IL-18 failed to rescue the susceptibility of Defa6-Cre⁺Stat3^{FF/FF} mice to AIEC infection by CFU assay (revised **Fig.4j**).

Together, these results provide direct genetic evidence that Paneth cells (result-1 above) or IL-18-Stat3 signaling in Paneth cells (result 2, 3 above) are crucial for host defense against AIEC.

5) Organoid growth and budding data in response to IL-22 and IL-18 stimulation (24 hours) is interesting. However, the cytokines dose (50ng/ml and 100ng/ml) seems very high and the treatment regime is limited to 24 hours given prior literature on the effect of continuous stimulation of IL-22 on organoid growth. The manuscript would benefit from additional description of the choice treatment time (24 hours) and dose. Could the author show that a lower dose of IL-18 has the same effect on organoid growth/budding?

Our response:

This is a good question to test. In our hands, to observe (and quantify) organoid growth and budding in a more reproducible manner, we found ileum organoids need to be cultured for at least 5 days. Therefore, we usually assay organoid growth and budding at day-5 with IL-22 and IL-18 given from day-0 to day-5 (i.e. continuous stimulation is needed to observe morphological changes, which are very difficult to detect with only 24h treatment). In contrast, for QPCR analysis of mRNA in IL-22 (or IL-18)-stimulated organoids, we need to use 24h treatment regime as continuous stimulation (day-0 to day-5) will give very high basal levels which we could not see the induction effect at all. To this reviewer's suggestion, we have titrated the IL-18 dose to test its effect on organoid growth and budding (revised **Fig.6f**). The results indicate the effects is IL-18 dose-dependent.

6) The authors should provide survival and weight curves as well as a histological score of the mice (IL22^{-/-}, IL18^{-/-} and Vil-cre;IL-22Ra1^{fl/fl}) infected with AIEC.

Our response:

This is a good question that we should provide more details in our manuscript. In brief, we are using a short and acute AIEC infection regime (six days) to ask how IL-22/IL-18 regulates Crohn's AIEC during host defense. In the original AIEC publication (Nat Comm 4:1957, 2013, we obtained the same AIEC NRG857c strain from this group), it takes up to 21 days to have detectable morphological changes and pathology scores in B6 mice (refer to **Fig.3**, Nat Comm 4:1957, 2013), where the authors have used it as a chronic inflammation model that is also indicated in the title of paper). Therefore, this AIEC clinical isolate is very much less pathogenic, compared to the popular *Citrobacter rodentium* (also an E coli-related strain) infection model. The only reason why we used this AIEC clinical isolate is to retain the clinical relevance of our study to Paneth cell/Crohn's disease in humans.

As such, it was true that all infected mice usually survived at day-6 (or even up to 3 weeks!) when we sacrificed them for the analysis and they showed no detectable pathology by H&E staining (WT vs IL-22/IL-18 KO in AIEC-d6, in revised **Fig.s4h,s8b**, only alterations of Paneth cell or Goblet cell numbers can be microscopically observed). We have updated this information in “Materials & Methods” to indicate this fact. In addition, we decide to remove the only *Vil-Cre⁺IL-22Ra1^{F/F}* mouse data (original **Fig.s1g**) as we unexpectedly lost this mouse strain (very bad breeding) during the revision. Instead, we provided new CFU data in AIEC-infected *Vil-Cre⁺Stat3^{F/F}* and *Defa6-Cre⁺Stat3^{F/F}* mice which are more relevant to our mechanistic study (revised **Fig.2i**).

7) Since IL-18 regulates *Lyz1* and *Cryptdin* expression at transcripts level, it would be useful to confirm the loss of Paneth cells in AIEC infected *Il18^{-/-}* mice by other methods (UEA1, alcian blue or phloxine-tatrazine).

Our response:

We thank the reviewer’s comments here. After literature search, we found that UEA1 can stain for Goblet cells and Paneth cells (Am J Vet Res 76:358, 2015). In one study, phloxine-tartrazine was used for Paneth cells while alcian blue was used for Goblet cells (Clin Exp Immunol 126:117, 2001). One recent study also used UEA1 for Goblet cells and Lysozyme for Paneth cells (Cell Reports 15:1743, 2016). In our experiments, we found Lysozyme⁺ Paneth cells were specifically ablated in Paneth cell-deficient mice (with *Defa6-Cre⁺Rosa-LSL-DTA*, revised **Fig.s1c**) and IL-22/IL-18-induced Paneth cells were compromised in stimulated *Defa6-Cre⁺Stat3^{F/F}* organoids (revised **Fig.s1f, s4d**). In addition, injection of IL-18 into *Defa6-Cre⁺Stat3^{F/F}* mice failed to fully promote Lysozyme⁺ Paneth cells (revised **Fig.4k-m**), indicating that Lysozyme is a very specific marker for Paneth cells (as also indicated in Development 139:488, 2012). In addition to Lysozyme/*Cryptdin* mRNA expression (revised **Fig.s1h, s4i**), we now provide immunofluorescence staining of ileum tissues for Lysozyme⁺ Paneth cells, in naïve vs AIEC-infected WT and IL-22 KO (revised **Fig.2h**) and IL-18 KO mice (revised **Fig.4f**).

8) The expression level of *Dll1*, *Dll4*, *Atoh1* and *Hes1* needs to be shown in *Il18^{-/-}* mice at steady state and after infection.

Our response:

We thank the reviewer’s comments here. *Dll1/Dll4* (Notch ligands), *Atoh1* (transcription factor for secretory cell differentiation), and *Hes1* (Notch target gene) are all key niche regulators between Paneth cells and stem cells (Cell Mol Gastroenterol Hepatol 7:255, 2019). Per this reviewer’s suggestion, we now provided the mRNA analysis of these markers in ileum crypts derived from naïve vs AIEC-infected IL-18 KO mice (revised **Fig.s4f**). The results show that most niche factors (*Atoh1* was not) were reduced in IL-18 KO crypts (naïve vs AIEC) that likely lead to impaired Paneth cells as well as stem cells.

Minor

- The major source of IL-22 is ILC3 in the intestine. This needs to be included in the introduction (second paragraph).

Our response:

We have updated the information in the “Introduction”, with related references cited (revised Ref-6, Ref-7).

- *Muc2* antibody source is missing

Our response:

We have updated the information in the “Materials & Methods”.

- What are the mechanisms of how IL-18R signaling promotes TCF4 nuclear translocation? If the author has any speculation, please discuss.

Our response:

We thank the reviewer’s comments here and we have provided more mechanism studies. IL-18, but not IL-22, specifically induces an Akt-TCF4-Lgr5 pathway to promote stem cells by inducing TCF4 binding to the P1 region within mouse *Lgr5* promoter (revised **Fig.7f-h**). We also discussed this findings in the result section.

- It would be useful to readers if the author includes discussion of their Paneth cell related findings with PMID30364840, PMID33060802, reference 22 and 24.

Our response:

We thank the reviewer's comments. We have cited and included a discussion (in introduction section) about these 4 references (revised Ref-16 to Ref-19).

(PMID 30364840, Cell Mol Gastroenterol Hepatol 7:1, 2019)

(PMID 33060802, Mucosal Immunol 14:389, 2021)

(Reference 22, Cell Mol Gastroenterol Hepatol 7:255, 2019)

(Reference 24, Nature 528:560, 2015)

- It is better to move figure 3C-D to supplemental.

Our response:

We thank the reviewer's comments. We agree that the original **Fig.3c-d** are IL-18-related results that have been reported in prior publications. To this end, we moved the original **Fig.3c** to revised **Fig.s2b** and the original **Fig.3d** to revised **Fig.s2d**, as suggested by this reviewer.

- It appears that IL-22-mediated STAT3-dependent organoid growth is independent of Claudin and Lgr5+ ISC. How does IL-22 promote organoid growth? If authors have any speculations, please discuss.

Our response:

We thank the reviewer's critical comments here. Based on gain-of-function and loss-of-function studies by us and others, a short answer is that IL-22 promotes Stat3-dependent Ki67⁺ epithelial proliferation (organoid growth) by targeting and upregulating transit-amplifying (TA) cells, but IL-22 suppresses Lgr5⁺ stem cells in a Stat3-independent (likely via inhibiting Wnt and Notch signaling) manner.

There are several evidence to support this statement.

(1) Reduced Ki67⁺ proliferating cells in IL-22^{-/-} crypts (revised **Fig.2b-c**). We found Ki67⁺ proliferating cells are predominantly located in "the TA compartments" in the middle and upper part of crypts but not stem cell compartments (enriched at the crypt base), as shown in revised **Fig.2b,6b**.

(2) IL-22 injection promotes Ki67⁺ crypts in vivo, which are predominantly located in the TA compartments (**Fig.6d**, Cell Mol Gastroenterol Hepatol 7:255, 2019). This reference is cited.

(3) IL-22 injection promotes crypt length (TA compartment) in vivo (**Fig.3d**, Nature 528:560, 2015) or crypt depth (TA compartment) in vivo (**Fig.6c-d**, Cell Mol Gastroenterol Hepatol 7:255, 2019). This reference is cited.

(4) In revised **Fig.6a**, IL-22-induced Ki67 mRNA is significantly reduced in Vil-Cre⁺Stat3^{F/F} organoids, which suggests that IL-22 promotes Ki67⁺ proliferating TA cells (with result-1 above) in a Stat3-dependent manner.

(5) Reduced Lgr5⁺ stem cells in both IL-22^{-/-} and Vil-Cre⁺Stat3^{F/F} crypts at the steady state and AIEC-d6 (revised **Fig.7a-b**). IL-22-induced organoid size is reduced in Vil-Cre⁺Stat3^{F/F} organoids compared to Stat3^{F/F} organoids (revised **Fig.6e**). This suggests that in vivo IL-22-Stat3 deficiency compromises TA cell proliferation which might adversely feedback to nearby Lgr5⁺ stem cells in homeostasis or during AIEC infection.

(6) The effect of IL-22 on organoid size is Claudin-2 dependent (**Fig.2**, Cell Mol Gastroenterol Hepatol 7:255, 2019) and we found that IL-22-induced Claudin-2 is Stat3-independent (revised **Fig.s6a**). However, IL-22 could induce activation of Stat1/3/5 (**Fig.1**, J Biol Chem 277:33676, 2002) and among them Stat1 has been shown to be critical for Claudin-2 expression (**Fig.3**, Ann NY Acad Sci 1405:116, 2017). Furthermore, both Stat6 (J Immunol 190:1849, 2013) and JNK (PLoS One 9:e85345, 2014) are involved in the induction of Claudin-2. Therefore, our result can only exclude a role of Stat3 in the induction of Claudin-2.

In brief, our results are consistent to the conclusion "IL-22 limits ISC expansion in favor of increased TA progenitor cell expansion" (Cell Mol Gastroenterol Hepatol 7:1, 2019). It appears that IL-22 can affect the mutual relationship between TA cells and nearby stem cells for a kinetics-balanced homeostasis. This is supported by the accumulating data (Cell Mol Gastroenterol Hepatol 7:1, 2019 and Cell Mol Gastroenterol Hepatol 7:255, 2019) and an Editorial comment "IL-22, and perhaps other cytokines, can play dual roles on neighboring cells types to modulate intestinal epithelial repair and barrier function" in Cell Mol Gastroenterol Hepatol 7:409, 2019 (same issue). We have provided some insights in the section of "Discussion".

Reviewer #3 (Remarks to the Author):

This is a potentially interesting study describing cooperativity between IL22 and IL18 during E coli infection. However, it is difficult to draw strong conclusions from the data presented. It is not convincing that the antimicrobial molecules attributed to Paneth cells are actually coming from Paneth cells. It is also unclear if changes in Paneth cell and Goblet cell frequencies are due to impacts on those cells or to impacts on their precursors, including ISCs. Please find below additional specific comments.

Major comments:

- Goblet cells can be a source of Reg3y, Ang4, and others. To confirm a direct effect of IL-18 and IL-22 on Paneth cells, it would be more convincing to treat sorted paneth cells with IL-22 or possibly to gavage Defa6-il22r1 f/f mice with AIEC and check for changes in antimicrobial activity.

Our response:

We thank the reviewer's critical comments here. In our hands, to sort Lysozyme⁺ Paneth cells, we need to perform intracellular staining of lysozyme (to label Paneth cells, we do not have a Lyz-reporter) which requires fixation of crypts. Consequently, fixed crypt cells can not be stimulated with IL-22 again for further analysis. Regarding the second approach, we unexpectedly lost Vil-Cre⁺IL-22Ra1^{F/F} mouse strain (very bad breeding) during the revision, so we could not set up breeding to obtain Defa6-Cre⁺IL-22Ra1^{F/F} mice. Because of this, we also decide to remove the only Vil-Cre⁺IL-22Ra1^{F/F} mouse data (original **Fig.s1g**). However, to address this reviewer's concern, we instead performed the following experiments, to confirm a direct effect of IL-22 and IL-18 on Paneth cells.

(1) Lysozyme is a very specific marker for Paneth cells (revised **Fig.s1c**) and Paneth cells contribute to AIEC infection (revised **Fig.1i**). Stat3 signaling in Paneth cells (by AIEC infection of Defa6-Cre⁺Stat3^{F/F} mice, revised **Fig.2i**) contributes to AIEC infection.

(2) IL-22-induced lysozyme (both protein and mRNA levels) was reduced in stimulated Defa6-Cre⁺Stat3^{F/F} organoids, compared to Stat3^{F/F} organoids (revised **Fig.2f, s1f-g**). This confirms a direct effect of IL-22 on Lysozyme⁺ Paneth cells.

(3) IL-18-induced lysozyme (both protein and mRNA levels) was reduced in stimulated Defa6-Cre⁺Stat3^{F/F} organoids, compared to Stat3^{F/F} organoids (revised **Fig.4b-c, s4d**). This confirms a direct effect of IL-18 on Lysozyme⁺ Paneth cells.

(4) IL-18 injection failed to fully rescue the susceptibility of Defa6-Cre⁺Stat3^{F/F} mice to AIEC infection by CFU assay (revised **Fig.4j**), indicative of a direct role of IL-18-Stat3 axis in Paneth cells during AIEC infection.

(5) IL-18 injection into Defa6-Cre⁺Stat3^{F/F} mice, compared to Stat3^{F/F} mice, showed reduced lysozyme levels (both protein and mRNA), indicative of a direct role of IL-18-Stat3 axis in promoting Lysozyme⁺ Paneth cells (revised **Fig.4k-m**).

- Authors suggest that IL-22 induces the upregulation of RegIII γ , Relmb, Itln1, and Ang4, all with anti-gram-negative. They also state that IL-18 can also induce Itln1 and Ang4. Because IL-18 is also induced by IL-22, they cannot exclude that Itln1 and Ang4 are not directly upregulated by IL-22 but indirectly through IL-18. To test this hypothesis they need to treat IL-18 KO organoids with IL-22 and check for gene changes. In the alternative, the fact that IL-18 can only induce some antimicrobial suggests that IL-22 and IL-18 may be targeting distinct cells types.

Our response:

This is a very good question to test. To address this, we analyzed organoid growth in IL-22-stimulated IL-18^{-/-} organoids and found that both budding and size of organoids were not affected by epithelial IL-18 deficiency (revised **Fig.6h**). However, certain IL-22-induced anti-microbial products (RegIII γ / Relm β / Muc2, which are non-Paneth cell-specific, and Lysozyme / Cryptdin / Ang4, which are Paneth cell-specific or -related) were partially compromised by epithelial IL-18 deficiency (revised **Fig.s5b, 8g**). This also suggests that IL-22 targets certain anti-microbial genes partially via epithelial IL-18. For in vivo rescue exp, injection of IL-22 into IL-18 KO mice failed to upregulate Lysozyme⁺ Paneth cells and rescue CFU or IFN γ during AIEC infection (revised **Fig.5d-f**). Injection of IL-22 into IL-18 KO mice failed to promote mucin secretion from Goblet cells during AIEC infection

(revised **Fig.8e**). Of note, IL-22/IL-18 show additive effect on bacteria-killing by fresh crypts (revised **Fig.4e**), indicating that they have some non-redundant role in crypt immunity. Alternatively, as pointed out by this reviewer, our results (revised **Fig.3g, s8a**) also suggest that IL-22R and IL-18R are differentially expressed (% or MFI) in distinct epithelial subsets that may affect the signal strength of IL-22R or IL-18R for gene induction. Taken together, these results indicate that IL-22 and IL-18 show some redundant and non-redundant effects, depending on the contexts which are being tested.

- Fig2 H, 2I. Data does not exclude the possibility that IL-22/STAT3 AXIS is acting on precursor secretory cells promoting differentiation rather than act on fully differentiated paneth cells. Is the reduction in lysozyme positive cells the results of increased apoptosis in adult paneth cells or reduced differentiation from precursors? Apoptotic markers, for example tunnel assay, together with lysozyme stain is required.

Our response:

This is a very good point. As suggested by this reviewer, we performed the following experiments.

(1) Deficiency of IL-22 or IL-18 indeed affect differentiated Paneth cells, evidenced by H&E staining and quantification of granule-containing Paneth cells per crypts (revised **Fig.s4h**).

(2) By TUNEL and Lysozyme staining in naïve vs AIEC-infected tissues, derived from WT vs IL-22 and IL-18 KO mice, Lysozyme⁺ Paneth cells are reduced and not apoptotic at the steady state in IL-22 or IL-18 KO ileum tissues (revised **Fig.s4j**, upper panels, Lysozyme⁺ Paneth cells are not TUNEL⁺). During AIEC infection, WT mice showed very few TUNEL⁺ cells in the tips of villus, while IL-22 KO showed significant TUNEL⁺ cells in the tips of villus. In contrast, IL-18 KO mice showed much more TUNEL⁺ cells in the transit-amplifying (TA) compartments and some in the villus tips (revised **Fig.s4j**, lower panels). The distribution of TUNEL⁺ crypts are not at the crypt base where Paneth cells are located, indicating that the loss of IL-22 or IL-18 might not contribute to apoptosis measured by TUNEL during AIEC infection.

(3) As immunofluorescence staining may not provide the best resolution at the single cell level, we analyzed active caspase-3 together with lysozyme staining by flow cytometry. The result showed that IL-18 deficiency indeed cause more apoptosis in the gated Paneth cells during AIEC infection (revised **Fig.s4k**).

(4) We also analyzed the mRNA levels of Atoh (a transcription factor required for secretory cell differentiation, Science 294:2155, 2001, Development 139:488, 2012) and Dll1/4 (Notch ligands in Paneth cells, Science 340:1190, 2013) in naïve vs AIEC-infected crypts of IL-18 KO mice (revised **Fig.s4f**). The results show that while IL-18 deficiency does not affect Atoh mRNA level, Dll1/Dll4 (Notch ligands) and Hes1 (Notch target gene) mRNA levels were reduced (Cell Mol Gastroenterol Hepatol 7:255, 2019), suggesting that altered niche factors between Paneth cells and stem cells, due to IL-18 deficiency, may affect the functionality or differentiation of Paneth cells from stem cells (or precursors).

(5) IL-22 or IL-18-induced organoid size was not affected in Defa6-Cre⁺Stat3^{F/F} organoids. However, IL-18 (but not IL-22)-induced organoid budding was compromised in Defa6-Cre⁺Stat3^{F/F} organoids (revised **Fig.s6b**). This is likely due to a negative feedback effect of niche (growth) factors from Paneth cells to support nearby stem cells. That is, functionality-compromised IL-18-stimulated Defa6-Cre⁺Stat3^{F/F} Paneth cells could not provide sufficient niche factors to fully support nearby stem cells for budding during culture.

Minor

- Fig 1F. The authors show a western blot with two bands but only one changes. Have the authors confirmed which band is LGR5?

Our response:

We thank the reviewer to point out this. Both bands are indeed Lgr5, as shown in Lgr5 knockout CH-212 neuroblast cells (**Fig.1f**, Sci Signal 13:waaz4051, 2020). However, to validate this, we knockdown (KD) Lgr5 in our CMT93 colon epithelial cells (revised **Fig.s1b**). While the upper band is moderately reduced, the lower band is completely abolished in two independent Lgr5^{KD} CMT93 clones, indicating that the lower band is authentic Lgr5. To fully convince the readers, where applicable, we always provide flow cytometry data for Lgr5 with a isotype control to accompany the WB data.

Below is a complete list of all Lgr5 WB data in the revised manuscript.

(1) Revised **Fig.1f** (WB data) with revised **Fig.1d** (flow cytometry data).

(2) Original **Fig.5d** (WB data, only naïve samples) was removed. Instead, we provide Lgr5 flow cytometry data (with isotype controls) using naïve vs AIEC-infected samples from IL-22 KO mice (revised **Fig.7a**) and from $Vil-Cre^{+}Stat3^{F/F}$ KO mice (revised **Fig.7b**).

(3) Original **Fig.s4e** (WB data) and original **Fig.s4f** (WB data) were removed, as we now provide IL-18-induced Lgr5 flow cytometry data with isotype controls in organoids or CMT93 cells (revised **Fig.7e,7g**) and WB data in CMT93 cells (revised **Fig.7g**).

(4) Revised **Fig.7g** have both WB data and flow cytometry data. The Akt inhibitor treatment clearly inhibits Lgr5 lower band in this case, correlated to flow cytometry data. This indicates that Lgr5 lower band in WB data is more functionally linked to Lgr5 stem cells.

- S1B why are crypts from WT mice so small and fragmented compared to Defa6-Cre⁺-Rosa mice?

Our response:

The quality of crypts depends on skillful sample processing during EDTA treatment/shaking which causes significant crypt death and damage. Because of this, we replaced the original **Fig.s1b** (fresh crypt image with granule-enriched Paneth cells) with the revised **Fig.s1c** (immunofluorescence staining for Lysozyme⁺ Paneth cells). The results clearly show the ablation of Lysozyme⁺ Paneth cells in Paneth cell-deficient (Defa6-Cre⁺Rosa-LSL-DTA) mice.

- Fig 2A: increase in mRNA for IL-22 is not always correlated with increased production and release of IL-22. At least ELISA should be done to test the amount of protein.

Our response:

We now provide both protein and mRNA measurements of IL-22 (revised **Fig.2a**) and IL-18 (revised **Fig.4a**).

- Fig 2B: staining of Ki67 excludes the nuclear region and is mainly in the cytoplasm, but normally Ki67 is a nuclear marker. Cells in the crypts regions have less Ki67 but the total number appears to be unchanged. Quantification of the average number of Ki67⁺ cells in each crypt should be performed.

Our response:

We thank the reviewer to point out this. Here we do not intend to quantify Ki67⁺ cells per crypts but would like to emphasize that the overall reduction of Ki67⁺ crypts is “in the TA compartments” of crypts in IL-22 KO mice at the steady state or during AIEC infection (immunofluorescence staining for Ki67 is clearly very distinct in WT vs KO, naïve vs AIEC). We also added “the AIEC-d6 data” in revised **Fig.2b** and the results indicate that IL-22 deficiency causes reduced crypt proliferation/regeneration, measured by Ki67⁺ cells. For quantification of Ki67⁺ cells, flow cytometry analysis of crypts for Ki67% crypts (revised **Fig.2c**) was performed, measured at the single-cell level. We provide the similar data in IL-18 KO samples (revised **Fig.6b-c**).

- Fig 2E and 2F: How do the authors explain the discrepancy between the in vitro and vivo findings where they observed the opposite results in absence of stat3? In homeostasis after depletion of stat3, *Itln1* and *Ang4* go up in organoids (Fig 2E) but go down (*Itln1*) or unchanged (*Ang4*) in ileum crypts (Fig S1F). In organoids, the depletion of Stat3 induces no changes in *Relmb*, but crypts show a dramatic reduction in *Relmb* after depletion.

Our response:

We thank the reviewer to point out this. While organoid culture is still widely-used and perhaps the best platform for in vitro mechanistic study in primary epithelial cells, it has major limitations because many growth factors are needed and added into the culture for optimal growth that consequently might have impacts on certain signaling pathways. For example, recombinant EGF is added to promote Ras-ERK signaling for optimal growth (Science 340:1190, 2013). Recombinant Noggin is added as a BMP-4 antagonist to restrict BMP-mediated stem cell suppression (Nat Comm 8:13824, 2017). Recombinant R-spondin 1 is added to provide enhanced Wnt signaling for stemness (EMBO J 31:2685, 2012), while Jagged-1 peptide (a Notch agonist) is used to enhance Notch signaling for stemness (J Exp Med 192:1365, 2000). Together these growth factors provide a very optimal

condition for organoid growth especially for stem cells but more or less they could create an intrinsic biased condition as well such that organoids are not “untouched” primary cells any more. Because of this, untreated organoids perhaps can not perfectly represent a homeostasis status and therefore we prefer to draw conclusions based on the effect of additional treatment such IL-22 or IL-18, compared to untreated conditions. With this being said, we believe that freshly-isolated crypts are a better source of “untouched” samples for comparison analysis. We have discussed this in our revised manuscript.

- (1) RegIII γ and Relm β mRNA basal levels are reduced in both IL-22^{-/-} crypts (revised **Fig.2d**) and Vil-Cre⁺Stat3^{F/F} crypts (revised **Fig.s1e**).
- (2) IL-22 fails to induce RegIII γ and Relm β mRNA in Vil-Cre⁺Stat3^{F/F} organoids (revised **Fig.2e**), but still could induce RegIII γ and Relm β mRNA in Defa6⁺Stat3^{F/F} organoids (revised **Fig.s1g**). This indicates that the major epithelial producer for RegIII γ and Relm β is not from Paneth cells.
- (3) At the steady state, Ang4 mRNA levels are not altered in both IL-22^{-/-} crypts (revised **Fig.2d**) and Vil-Cre⁺Stat3^{F/F} crypts (revised **Fig.s1e**).
- (4) IL-22 stimulation still promotes Ang4 mRNA in Vil-Cre⁺Stat3^{F/F} organoids (revised **Fig.2e**), suggesting that IL-22-induced Ang4 is Stat3-independent in certain epithelial subset. Note that there is a much higher basal level (untreated) of Ang4 in Vil⁺Stat3^{F/F} organoids compared to Stat3^{F/F} organoids, suggesting there might be an effect of the optimized culture condition on Ang4 gene regulation.
- (5) IL-22 stimulation fails to promote Ang4 mRNA in Defa6⁺Stat3^{F/F} organoids (revised **Fig.s1g**), suggesting that IL-22-induced Ang4 in Paneth cells is Stat3-dependent. As noted, there is no difference in the basal level (untreated) of Ang4 between Stat3^{F/F} vs Defa6⁺Stat3^{F/F} organoids. This indicates that Ang4 could be induced by non-Paneth cells such as Goblet cells as mentioned by this reviewer.
- (6) IL-22 stimulation fails to promote Itln1, Lysozyme, and Cryptdin mRNA in Vil-Cre⁺Stat3^{F/F} organoids (revised **Fig.2e**), as well as in Defa6⁺Stat3^{F/F} organoids (revised **Fig.s1g**).
- (7) As noted in revised **Fig.s1c**, Lysozyme represents a very specific marker for Paneth cells as there is completely no signals in Paneth cell-deficient mice (with Defa6-Cre⁺Rosa-LSL-DTA mice).

- Fig 4G -4F: FACS data and immunofluorescence suggest that depletion of IL-18 alone is sufficient for changes in paneth cell numbers. Can the loss of paneth cells be rescued in IL-18 -/- mice by injecting IL-22 after AIEC gavage?

Our response:

We thank the reviewer to point out this, which we also consider as an important mechanistic data that is missing in our pervious manuscript. To address this, we have performed three stes of rescue experiments to dissect whether IL-22 regulates Paneth cells via IL-18 (or IL-18 dependent).

Rescue Exp-1 (revised **Fig.5a-c**, data shown are combined results from two independent exp):

In revised **Fig.5a**, IL-18 injection into IL-22 KO mice indeed rescue the susceptibility of IL-22 KO to AIEC, by reducing CFU in most intestinal samples to those levels of WT mice and by increasing IFN γ ⁺ CD4⁺ T (%+MFI) and CD8⁺ T (%) cells in the lamina propria (revised **Fig.5b**). Furthermore, in revised **Fig.5c**, while IL-18 injection into WT mice did not further upregulated lysozyme during AIEC infection, injection of IL-18 into IL-22 KO mice significantly promotes lysozyme, which is a very specific marker of Paneth cells. Upregulation of IFN γ ⁺ T cells and lysozyme levels in IL-18-injected IL-22 KO mice provides in vivo evidence that IL-22 could regulate IFN γ via IL-18, or support a direct link of IL-22 - IL-18 - IFN γ axis during AIEC infection.

Rescue Exp-2 (revised **Fig.5d-f**, data shown are representative results from two independent exp):

To further support the above conclusion, we performed a reciprocal rescue exp to test whether IL-22 injection into IL-18 KO mice could rescue the susceptibility of IL-18 KO to AIEC. As noted in revised **Fig.5d**, IL-22 injection into IL-18 KO mice failed to reduce CFU in most intestinal samples to those levels of WT mice. Correlated to this, IL-22 injection also failed to upregulate IFN γ ⁺ CD4⁺ T and CD8⁺ T cells in the lamina propria (revised **Fig.5e**). Furthermore, in revised **Fig.5f**, injection of IL-22 into IL-18 KO mice clearly can not upregulate lysozyme levels in IL-18 KO. Together, these results further support a direct link of IL-22 - IL-18 - IFN γ axis, where IL-22

is a bona fide upstream regulator of IL-18 leading to IFN γ production in T cells. These results also indicate a dominant requirement for IL-18 over IL-22 in Paneth cell numbers and function during AIEC infection.

Rescue Exp-3 (revised **Fig.4j~4m, s4l**, data shown are combined results from two independent exp):

To test the key role of IL-18 in Lysozyme⁺ Paneth cells in vivo and to further support the above two rescue exp (revised **Fig.5c** and **5f**), we performed the third rescue exp to test whether IL-18 injection into Defa6-Cre⁺Stat3^{F/F} mice could rescue the susceptibility of Paneth cell-specific Stat3 conditional KO mice to AIEC infection. As noted in revised **Fig.4j**, IL-18 injection into Defa6-Cre⁺Stat3^{F/F} mice failed to reduce CFU in all intestinal samples to those levels of WT mice. Injection of IL-18 into AIEC-infected Defa6-Cre⁺Stat3^{F/F} mice also failed to upregulate lysozyme⁺ Paneth cells (revised **Fig.4k-i**) and mRNA levels of Paneth cell-related products (Lysozyme/Cryptdin in revised **Fig.4m** and Itln1/Ang4 in revised **Fig.s4l**).

In brief, to address this reviewer's question, injection of IL-22 into IL-18 KO mice during AIEC infection could not rescue Lysozyme⁺ Paneth cells. Similarly, injection of IL-22 into IL-18 KO mice failed to promote mucin secretion from Goblet cells during AIEC infection (revised **Fig.8e**).

- Evaluation for evidence of direct phosphorylation of STAT3 in Paneth cells by IL-22 and IL-18 is necessary. In addition, it would be useful to evaluate the total amount of p-stat3 in goblet cells after treatment with IL-22 and IL-18.

Our response: We thank the reviewer to point out this. We performed several experiments to address this question. In brief, we found that while IL-22-induced phospho-Stat3 is relatively more stable or sustaining for detection, IL-18-induced phospho-Stat3 is much weaker and appeared transient for detection. The signal strength of phospho-Stat3 in in vitro-stimulated fresh crypts (see the crypt results below) is relatively weak and is undetectable in stimulated organoids (also reported in Extended Data **Fig.7** in Nature 528:560, 2015), or undetectable by immunofluorescence staining of tissue section (not shown). The only working condition for us is to use fresh crypts isolated from IL-22- or IL-18-injected mice with flow cytometry analysis of pStat3 in the gated epithelial subsets. In revised **Fig.4d**, we provided the results of phosphorylation of Stat3 by IL-22 and IL-18 in the gated Lysozyme⁺ Paneth cells and Muc2⁺ Goblet cells using fresh crypts isolated from IL-22- or IL-18-injected mice.

REVIEWER COMMENTS

Reviewer #1 (Remarks to the Author):

The authors have successfully addressed all my comments and concerns

Reviewer #2 (Remarks to the Author):

The authors have addressed all my prior comments and the manuscript is significantly improved. Overall, this outstanding work addressed an important gap in knowledge of IL-22-IL-18 crosstalk in the small intestine.

I still have some minor concerns/suggestions related to the new data.

- 1) IL-18 injection reduced bacterial burden in control Stat3-floxed mice as shown in figure 4j. However, in another experiment (figure 5a) IL-18 administration failed to reduce the bacteria burden in feces, jejunum and colon of WT mice. The authors have any speculation please discuss.
- 2) In figure 5d, the authors showed that IL-22 injection into Il18^{-/-} mice failed to improve bacterial clearance. However, IL-22 administration does not have any effect on WT mice bacteria clearance as well. This may impact the conclusion. Any speculation please discuss.
- 3) It appears that the control group (untreated) used in the flow plot of figure 2f and 4c are the same. Please correct. Do CMT93 cells express Lgr5 or a transfected cell line was used in figure 7g? It is not clear whether littermate control mice used in 8a and 8d.

Reviewer #3 (Remarks to the Author):

The manuscript is improved by the inclusion of new experiments, although the conclusion of direct effects of the IL-22/IL-18 axis on Paneth cells remain overstated. Paneth cells were not purified to examine direct effects, and use of transgenic mice lacking signaling molecules within Paneth cells does not prove what may be upstream. Additionally, while the authors state that they could not purify Paneth cells because of the lack of a reporter, cell surface phenotypes have been described that can be used to purify Paneth cells (see for example Rothenberg et al. Gastroenterology 2012. Identification of a cKit(+) colonic crypt base secretory cell that supports Lgr5(+) stem cells in mice. PMID: 22333952 PMCID: PMC3911891). The authors have clearly put a lot of work into this study, which provides a meaningful contribution to the literature. The above issues could be reconciled by modifying the conclusions and interpretations to be more modest, stating only that Paneth cells are impacted, and not directly targeted.

This is the rebuttal letter for the manuscript (NCOMMS-20-42875A), “**IL-22 initiates an IL-18 epithelial response circuit for host defense boost**”. Our point-to-point response (shown in black color) to each concern raised by reviewers (shown in green color) is listed below.

Reviewer-2 (Remarks to the Author):

The authors have addressed all my prior comments and the manuscript is significantly improved. Overall, this outstanding work addressed an important gap in knowledge of IL-22-IL-18 crosstalk in the small intestine. I still have some minor concerns/suggestions related to the new data.

1) IL-18 injection reduced bacterial burden in control Stat3-floxed mice as shown in figure 4j. However, in another experiment (figure 5a) IL-18 administration failed to reduce the bacteria burden in feces, jejunum and colon of WT mice. The authors have any speculation please discuss.

Our response:

(a) We thank the reviewer’s comments on this discrepancy between two animal studies. As the original **Fig.4j** and **Fig.5a** are combined results from two independent experiments, they have intrinsic limitations that the outcome of CFU assays could be affected by many variables when individual experiment was independently performed, including AIEC dose (we tried to perform several experiments in a short period of time to make sure AIEC titering was well-controlled), mouse age/gender, housing conditions, and techniques. Specifically, two independent experiments in the original **Fig.4j** gave almost the same results that we could easily combine the data. For the original **Fig.5a**, the rescue of bacterial clearance in IL-22^{-/-} mice by IL-18 injection was reproducible in two independent experiments, but only the 2nd experiment showed that IL-18 injection promotes AIEC clearance in WT mice (see 1st and 2nd Exp in **Fig.5a** below).

Considering the key purpose of this experiment (**Fig.5a**) is to test the effect of IL-18 injection in IL-22^{-/-} mice, we chose to show combined data to increase the statistical significance. However, by doing so, it leads to no effect of IL-18 injection in WT mice as noted. Correlated to this, the reviewer might notice that there are more variations in WT CFU assays in original **Fig.5a**, compared to original **Fig.4j**. Similar trend can also be observed in original **Fig.5b**, where combined data showed no effect of IL-18 injection to promote IFN_γ response in WT mice (but yes in the IL-22^{-/-} mouse group). Therefore, based on the sample size of original **Fig.5a-5c** (they were all paired experiments so the analyses are challenging), we are still confident to conclude that IL-18 injection indeed can rescue defective AIEC clearance in IL-22^{-/-} mice by promoting T-cell IFN_γ and epithelial Lysozyme responses

(also please note that the condition to achieve statistical significance by Two-way ANOVA is more strict, particularly when 1st and 2nd Exp, each of which has different output response level, were combined), but may not have a clear promoting effect on AIEC CFU, IFN γ response, or Lysozyme response in WT mice.

(b) Another contributing factor to the discrepancy of the original **Fig.4j** and **Fig.5a** could be microbiota. Because the littermate mice were used and even the housing conditions were the same, we could not exclude the possibility that the microbiota is different in our Stat3^{F/F} (**Fig.4j**) vs WT (**Fig.5a**) mice, which may affect the rescue effect of IL-18 injection. Furthermore, due to the limited timeframe, as well as littermate mice and huge amount of recombinant IL-18 needed for **Fig.4j** and **Fig.5a**, our experimental output is somewhat limited and in vivo data can not be perfect, unfortunately.

2) In figure 5d, the authors showed that IL-22 injection into Il18^{-/-} mice failed to improve bacterial clearance. However, IL-22 administration does not have any effect on WT mice bacteria clearance as well. This may impact the conclusion. Any speculation please discuss.

Our response:

(a) We thank the reviewer's comments here as this may raise a concern that our recombinant IL-22 has no biological effect, which indeed could impact the conclusion. For rescue experiments, we purchased several bulk quantities of recombinant IL-22 from Biolegend (#576206, all have the same batch numbers), which include an activity test, as determined by a dose dependent stimulation of human Colo205 cells in production of IL-10. With this being said, we used the same recombinant IL-22 in the same injection protocol (revised Supplementary **Fig.s6a**) to test in vivo Stat3 phosphorylation (revised **Fig.4d**, Supplementary **Fig.s4e**) and Mucin secretion (revised **Fig.8c**) in WT mice during AIEC infection. Both results show that our recombinant IL-22 from the same bulk batch has biological effect when injected into WT mice at the day-6 post AIEC infection. This confirms that the experiments in revised **Fig.5d-f** were performed in the working injection protocol (revised Supplementary **Fig.s6a**) with working recombinant IL-22 (revised **Fig.4d**, **Fig.8c**).

(b) As noted in WT mice (revised **Fig.5d**), endogenous IL-22 levels during AIEC infection appear to be sufficient for a protective effect which can not be further enhanced by more exogenous IL-22. In a similar case, the induction of more IL-22 in WT mice also fails to enhance the protection in an animal model of DSS-induced colitis (Fig4. J Immunol 196:34, 2016). Being a cytokine mainly derived from very limited innate cells like ILC3, it is conceivable that the total level of IL-22 may not be crucial but the very local concentration of IL-22 in tissues and its signaling effect is the key driver for initiating and boosting immune response at the early stage of infection.

(c) The main conclusion of **Fig.5** is that IL-18 functions downstream of IL-22 and relays IL-22 signaling to induce Paneth-cell Lysozyme and T-cell-derived IFN γ . As such, the missing IL-18 signaling components in IL-18^{-/-} mice can not be rescued by IL-22 injection (**Fig.5d-f**), but the missing IL-22 signaling components in IL-22^{-/-} mice instead can be effectively rescued by IL-18 injection (**Fig.5a-c**). However, why IL-22 injection failed to promote enhanced host defense (CFU and IFN γ response) in WT mice? Another possibility could be the limiting levels of IL-12 in WT mice. That is, while IL-22 injection may promote more IL-18 production in WT mice, however without more IL-12, the IFN γ response (which needs IL-12+IL-18 as shown in **Fig.s2b**) may not be further promoted in WT mice to enhance bacterial clearance. In **Fig.4j**, why IL-18 works in WT mice to further enhance AIEC clearance? In addition to Stat3^{F/F} (not pure WT as used in **Fig.5d**) mice being used, IL-18 might upregulate IL-12 locally, so more IL-18 (and more IL-12) leads to enhanced IFN γ response and better AIEC clearance. Intriguingly, it is reported that IL-12 upregulates IL-18R expression in T cells (J Immunol 167:1306, 2001) and that IL-12/IL-18 also upregulate IL-12R β 1 in macrophages (J Leukoc Biol 68:707, 2000), so the interplay of IL-18 and IL-12 could be evident in WT mice but not the interplay of IL-18 and IL-22.

3) It appears that the control group (untreated) used in the flow plot of figure 2f and 4c are the same. Please correct. Do CMT93 cells express Lgr5 or a transfected cell line was used in figure 7g? It is not clear whether littermate control mice used in 8a and 8d.

Our response:

(a) The original **Fig.2f** and **Fig.4c** were actually performed in the same experiment so they shared the same control groups. Due to the writing flow of the manuscript, we chose to split the results to mention the IL-22 part (Fig.2f) first, and the IL-18 part (Fig.4c) later. As we have two independent experiments, we have revised the results and used different control groups in Fig.2f vs Fig.4c. We also indicate this fact in the figure legends of Fig.2 and Fig.4 accordingly.

(b) CMT93 cells do express Lgr5 as shown in the original **Fig.7g**. We did not use a Lgr5-transfected CMT93 cell line here.

(c) Due to the limited supply of littermate animals during the 1st revision (most littermate mice were used in those 6x rescue experiments), we did not use appropriate littermate controls in the original **Fig.8a** and **Fig.8d**. To correct this, we have performed some new experiments and update the following data, where applicable throughout the whole manuscript, all with appropriate littermate controls, including the revised **Fig.8a** (original Fig.8a), revised **Fig.8b** (original Fig.8d), revised **Fig.s9a** (original Fig.s8a), revised **Fig.s9c** (original Fig.s8b), revised **Fig.s9d** (original Fig.s8c), revised **Fig.s2f** (original Fig.s2f), revised **Fig.s4h** (original Fig.s4h), and revised **Fig.s4j** (original Fig.s4j). Due to changes in Fig.8, the original **Fig.8b-c** were combined and moved to revised **Fig.s9b**. The original **Fig.8g** was move to revised **Fig.s9e**.

Reviewer-3 (Remarks to the Author):

The manuscript is improved by the inclusion of new experiments, although the conclusion of direct effects of the IL-22/IL-18 axis on Paneth cells remain overstated. Paneth cells were not purified to examine direct effects, and use of transgenic mice lacking signaling molecules within Paneth cells does not prove what may be upstream. Additionally, while the authors state that they could not purify Paneth cells because of the lack of a reporter, cell surface phenotypes have been described that can be used to purify Paneth cells (see for example Rothenberg et al. *Gastroenterology* 2012. Identification of a cKit(+) colonic crypt base secretory cell that supports Lgr5(+) stem cells in mice. PMID: 22333952 PMID: PMC3911891). The authors have clearly put a lot of work into this study, which provides a meaningful contribution to the literature. The above issues could be reconciled by modifying the conclusions and interpretations to be more modest, stating only that Paneth cells are impacted, and not directly targeted.

Our response:

(a) We very much thank the reviewer's comments. We agree with this reviewer's suggestion that Paneth cells can be sorted by using certain surface markers (SSC^{high} CD24⁺ CD66a^{-low} CD44⁺ c-Kit⁺), as reported by several recent papers (the first one is referred by the Reviewer) listed below.

Gastroenterol 142:1195, 2012 (Fig.2, Colonic Paneth cell-equivalent cells sorted for transcriptome analysis).
FASEB J 34:10299, 2019 (Fig.5, CD24⁺SSC^{high} Paneth cells were sorted and co-cultured with Lgr5⁺ organoids).
PNAS 116:26599, 2019 (Fig.5, CD24^{high}SSC⁺ Paneth cells were sorted for single-cell transcriptome analysis).
Nat Comm 12:3339, 2021 (Fig.2, Paneth cells were sorted for mRNA analysis).

However, we found in all studies, sorted Paneth cells (or the equivalent cell type in the colon) were used "immediately" for mRNA/transcriptome analysis but not cultured/stimulated for functional assays. The only paper (Fig.5, *FASEB J* 34:10299, 2019) used sorted Paneth cells to co-culture with Lgr5⁺ organoids, but again their readout is to evaluate organoid formation but not Paneth cells themselves. In parallel experiments (revised **Fig.s5c**, combined data from two independent exp), we found epithelial cells (unless organoids are used) die rapidly during culture/stimulation, which is particularly true when gut fragments or fresh crypts are used for culture/stimulation. We suspect sorted Paneth cells may not be in a good condition during long-hour culture so we could not achieve the similar induction results obtained by using organoids.

While we do not want to culture cells too long in order to avoid cell death, 8hr stimulation is minimum needed to see a good mRNA induction by IL-18. With these concerns, we worked with our Sorting Facility and based on their sorting capacity and some failed pilot experiments, we could only sort ileum crypts (3x mice /2hr sorting x 3 sorting timeframes = total 9 mice /6hr sorting per day, 3x mice were killed for fresh crypts immediately before each sorting timeframe) to purify sufficient amount of Paneth cells (about 150,000 cells, 75,000 for untreated and 75,000 for IL-18 stimulation) for one culture/stimulation in a day. We have performed three independent experiments (total 27x mice) and fortunately we could combine all data to show statistical significance (revised **Fig.s5a-c**). We hope our new experiments can support our conclusion that "IL-18 can target and stimulate Paneth cells directly".

(b) Based on the sorting strategy in *Gastroenterol* 142:1195, 2012 (colons were used), we tested by using ileum crypts. Different from colonic samples, we found both c-Kit⁺ (Q1) and c-Kit⁻ (Q4) cells within CD45⁻ EpCAM⁺ SSC⁺ CD24^{+high} CD66a^{-low} populations contain Lysozyme⁺ Paneth cells (revised **Fig.s5a**). To be consistent and also referable to the publication (*Gastroenterol* 142:1195, 2012), we chose to only sort CD45⁻ EpCAM⁺ SSC⁺ CD24^{+high} c-Kit⁺ CD66a^{-low} cell population to be used as purified Paneth cells (which is comprised of only 0.27% of total ileum crypts) for our culture/stimulation experiments.

(c) Per this reviewer's suggestion, we also modified some of our conclusions to be more modest.

On Page-4, "direct" was removed

To further validate a **direct** regulatory role for IL-22 in Paneth cell function, loss-of-function studies by flow cytometry, immunofluorescence and quantitative PCR analyses of Paneth cells in ileum tissues from steady-state or AIEC-infected IL-22-deficient mice were tested (**Fig.2g-h**, and Supplementary **Fig.s1h**).

On Page-8, "targeting" was removed

This result argues that it is actually IL-22-induced epithelial IL-18 that is **targeting** and upregulating Paneth cells for AIEC host defense.

On Page-8

"upregulate" is changed to "impact"

Taken together, our genetic evidence show that during AIEC infection, lamina propria-derived IL-22 and IL-22-induced epithelial IL-18 are functionally linked and coordinated to **impact** anti-microbial innate immunity of Paneth cells and subsequent adaptive IFN γ response in T cells.

On Page-11 "targeting" was removed

Again, this result argues that it is actually IL-22-induced epithelial IL-18 that is **targeting** and upregulating Goblet cells for AIEC host defense.

REVIEWERS' COMMENTS

Reviewer #2 (Remarks to the Author):

All comments have been sufficiently addressed.

This is our response letter for the manuscript (NCOMMS-20-42875B), “**IL-22 initiates an IL-18 epithelial response circuit for host defense boost**”. Our point-to-point response (shown in black color) to each concern raised by reviewers (shown in green color) is listed below.

Reviewer #1 (Remarks to the Author):

The authors have successfully addressed all my comments and concerns.

Our response:

We thank this reviewer’s satisfaction with our revised manuscript.

Reviewer #2 (Remarks to the Author):

All comments have been sufficiently addressed.

Our response:

We thank this reviewer’s satisfaction with our revised manuscript.

Reviewer #3 (Remarks to the Author):

All comments have been sufficiently addressed.

Our response:

We thank this reviewer’s satisfaction with our revised manuscript.